# TRAINING ON TEST PROTEINS IMPROVES FITNESS, STRUCTURE, AND FUNCTION PREDICTION

## ABSTRACT

Data scarcity and distribution shifts often hinder the ability of machine learning models to generalize when applied to proteins and other biological data. Self-supervised pre-training on large datasets is a common method to enhance generalization. However, striving to perform well on all possible proteins can limit model's capacity to excel on any specific one, even though practitioners are often most interested in accurate predictions for the individual protein they study. To address this limitation, we propose an orthogonal approach to achieve generalization. Building on the prevalence of self-supervised pre-training, we introduce a method for self-supervised fine-tuning at test time, allowing models to adapt to the test protein of interest on the fly and without requiring any additional data. We study our test-time training (TTT) method through the lens of perplexity minimization and show that it consistently enhances generalization across different models, their scales, and datasets. Notably, our method leads to new state-of-the-art results on the standard benchmark for protein fitness prediction, improves protein structure prediction for challenging targets, and enhances function prediction accuracy.

## 1 INTRODUCTION

A comprehensive understanding of protein structure, function, and fitness is essential for advancing research in the life sciences (Subramaniam & Kleywegt, 2022; Tyers & Mann, 2003; Papkou et al., 2023). While machine learning models have demonstrated remarkable potential in protein research, they are typically optimized for achieving the best average performance across large datasets (Jumper et al., 2021; Watson et al., 2023; Yang et al., 2024; Kouba et al., 2023). However, biologists often focus their research on individual proteins or protein complexes involved for example in metabolic disorders (Ashcroft et al., 2023; Gunn & Neher, 2023), oncogenic signalling (Hoxhaj & Manning, 2020; Keckesova et al., 2017), neurode-generation (Gulen et al., 2023; oh Seo et al., 2023), and other biological phenomena (Gu et al., 2022). In these scenarios, detailed insights into a single protein can lead to significant scientific advances.

Nonetheless, general machine learning models for proteins often struggle to generalize to individual case studies due to data scarcity (Bushuiev et al., 2023; Chen & Gong,

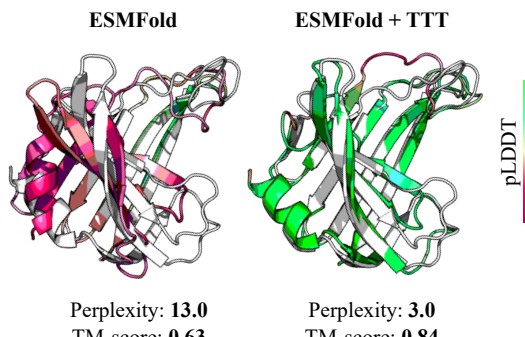

**ESMFold**     **ESMFold + TTT**

Perplexity: **13.0**     Perplexity: **3.0**
TM-score: **0.63**     TM-score: **0.84**

Figure 1: **Example of test-time training (TTT) applied to protein folding.** ESMFold poorly predicts the structure of the CASP14 target T1074 (shown in white) because the underlying language model ESM2 poorly fits the sequence, as indicated by the high perplexity (Fig. 2E in Lin et al. (2023) and the left panel here). Self-supervised test-time training of ESM2 on the single sequence of T1074 minimizes the perplexity, leading to improved structure prediction (better TM-score alignment and higher pLDDT predicted confidence). The same test-time training approach is also broadly applicable to other tasks, such as protein fitness and function prediction.

2022) and distribution shifts (Tagasovska et al., 2024; Feng et al., 2024). Bridging the gap between broad, dataset-wide optimizations and the precision required for studying single proteins in

practical applications remains a critical challenge in integrating machine learning into biological research (Sapoval et al., 2022).

By contrast, in other application domains of machine learning, such as computer vision and natural language processing, customization and adaptation approaches have emerged as powerful tools to improve model performance in specific contexts (Ruiz et al., 2023; Hardt & Sun, 2023). Drawing inspiration from the test-time training (TTT) approach developed in computer vision to mitigate distribution shifts (Sun et al., 2020; Gandelsman et al., 2022), in this work we propose the TTT approach for proteins. Our method enables adapting protein models to one protein at a time, on the fly, and without the need for additional data. Given a model that has been pre-trained using masked language modeling, our method minimizes the perplexity of the model on a given test protein through self-supervised fine-tuning, which, in turn, results in improved downstream performance without updating the downstream task head.

The prevalence of masked modeling in protein machine learning makes our method broadly applicable to various downstream tasks. Empirically, we demonstrate its effectiveness across three key challenges in protein machine learning. First, TTT achieves state-of-the-art results on the ProteinGym dataset (Notin et al., 2024), a well-established benchmark for protein fitness prediction. Second, TTT enhances protein structure predictions with ESMFold (Lin et al., 2023) and ESM3 (Hayes et al., 2024) on challenging targets. Third, the application of TTT to protein function predictors results in improved classification of terpene synthase (TPS) substrates and protein subcellular localization.

In summary, the key contributions of this work are three-fold:

1. Motivated by the generalization challenges and distribution shifts prevalent in protein machine learning, we introduce a new test-time training (TTT) method[1] that enables models to adapt to individual proteins on the fly and without requiring additional data.

2. We establish a link between our TTT approach and perplexity minimization, providing an insight into why this approach enhances model effectiveness.

3. We empirically validate TTT, achieving state-of-the-art results in protein fitness prediction, improving the protein structure prediction capabilities of well-established folding models, and enhancing protein function predictions in the tasks of terpene synthase substrate classification and protein localization prediction.

## 2 BACKGROUND AND RELATED WORK

In this section, we present the context and related work that highlight the rationale, feasibility, and broad applicability of test-time training (TTT) in the domain of machine learning on proteins. The widespread adoption of Y-shaped architectures relying on masked modeling enables the development of a general method for adapting protein models at test time via masking-based self-supervised fine-tuning.

**The Y-shaped paradigm of learning.** In machine learning applied to biology, architectures often follow a Y-shaped paradigm (Gandelsman et al., 2022), consisting of a backbone feature extractor $f$, a self-supervised head $g$, and an alternative fine-tuning head $h$. During training, $g \circ f$ is first pre-trained, after which the pre-trained backbone $f$ is reused to fine-tune $h \circ f$ toward a downstream task. Here, $\circ$ denotes a composition of two machine learning modules (e.g., $g$ is applied on top of $f$ in $g \circ f$). At test time, the final model $h \circ f$ is fixed. Generalization is achieved by leveraging the rich knowledge encoded in the backbone $f$ and the task-specific priors acquired in the fine-tuning head $h$. This paradigm enables overcoming data scarcity during fine-tuning and underlies breakthrough approaches in protein structure prediction (Lin et al., 2023), protein design (Watson et al., 2023), protein function prediction (Yu et al., 2023), and other protein-related tasks (Hayes et al., 2024).

The backbone $f$ is typically a large neural network pre-trained in a self-supervised way on a large dataset using a smaller pre-training projection head $g$ (Hayes et al., 2024). The fine-tuning head $h$, however, depends on the application. In some cases, $h$ is a large neural network, repurposing the pre-trained model entirely (Watson et al., 2023; Lin et al., 2023); in others, $h$ is a minimal projection

---

[1]https://github.com/anton-bushuiev/ProteinTTT

with few parameters (Cheng et al., 2023), or even without any parameters at all (i.e., a zero-shot setup, (Meier et al., 2021; Dutton et al., 2024)). In some cases, the fine-tuning head $h$ may also be a machine learning algorithm other than a neural network (Samusevich et al., 2024).

**Masked modeling.** While the objective of fine-tuning $h \circ f$ is determined by the downstream application, the choice of pre-training objective for $g \circ f$ is less straightforward. Nevertheless, most methods employ various forms of masked modeling, i.e., optimizing the model weights to accurately reconstruct missing parts of proteins, regardless of the downstream application. Masked modeling pre-traning underpins models for protein structure (Lin et al., 2023) and function (Samusevich et al., 2024) prediction, as well as for protein design (Hayes et al., 2024). For example, in AlphaFold2, a significant part of the loss function weight is put onto masked modeling of multiple sequence alignments (MSAs) (Jumper et al., 2021), and the model has been effectively fine-tuned for various tasks beyond structure prediction (Jing et al., 2024; Cheng et al., 2023; Motmaen et al., 2023).

Masked modeling is a dominant pre-training objective not only across different tasks but also across various protein representations. Sequence models applied to proteins are typically pre-trained to predict randomly masked amino acids in a random or autoregressive manner (Lin et al., 2023; Rao et al., 2021; Elnaggar et al., 2023; Madani et al., 2023; Ferruz et al., 2022; Rives et al., 2021; Rao et al., 2020). Models utilizing graph neural networks or 3D convolutions on protein structures are also commonly pre-trained to fill in missing structural fragments (Dieckhaus et al., 2024; Diaz et al., 2023; Bushuiev et al., 2023; Hsu et al., 2022; Shroff et al., 2020). The most recent approaches combine both sequential and structural information under masked modeling (Hayes et al., 2024; Su et al., 2023; Heinzinger et al., 2023).

**Model adaptation.** In many scenarios, machine learning models for proteins benefit from being adapted to a specific protein of interest. This adaptation is commonly achieved in two ways: either via additional input features or via protein-specific fine-tuning. Multiple sequence alignments (MSAs) containing sequences similar to the target protein provide a common way of supplying a model with protein-specific features (Abramson et al., 2024; Jumper et al., 2021; Rao et al., 2021). Another approach for injecting protein-specific knowledge into the model is standard supervised fine-tuning (i.e., via the $h \circ f$ track) on protein-specific data (Notin et al., 2024; Kirjner et al., 2023; Rao et al., 2019). An alternative is self-supervised fine-tuning (i.e., via the $g \circ f$ track) on proteins from the MSA (Notin et al., 2022b; Frazer et al., 2021; Alley et al., 2019) or on proteins sharing another property with the target protein, such as common family (Sevgen et al., 2023) or class (Samusevich et al., 2024). However, constructing MSAs is time-consuming (Fang et al., 2023), and similar proteins may not be available for many targets (Durairaj et al., 2023; Lin et al., 2023).

Here, we propose an extreme case of self-supervised fine-tuning: learning from a single target protein, without the need for any additional data. To the best of our knowledge, this approach has not been employed in the field of machine learning applied to biology; however, similar methods have been developed in computer vision (Chi et al., 2024; Wang et al., 2023; Xiao et al., 2022; Karani et al., 2021) and natural language processing (Hardt & Sun, 2023; Ben-David et al., 2022; Banerjee et al., 2021). The paradigm of test-time training (TTT), developed to mitigate distribution shifts in computer vision applications (Gandelsman et al., 2022; Sun et al., 2020), is a main inspiration for our work. Here, we demonstrate that TTT is highly relevant for machine learning on proteins even without the presence of explicit distribution shift. We investigate the link of TTT to perplexity minimization and show that TTT improves performance on several downstream tasks.

## 3 Test-time training (TTT) on proteins

As discussed in the previous section, many machine learning models for proteins employ Y-shaped architectures, consisting of a backbone $f$ with a self-supervised head $g$ and a supervised head $h$. This design facilitates the use of self-supervised fine-tuning across various tasks and models. Notably, most of these models leverage masked modeling as a pre-training objective, which enables the introduction of a broadly applicable test-time training (TTT) method based on masking. Our method adapts models to specific test proteins through masked modeling (Figure 2). In this section, we first formally define the proposed TTT approach (Section 3.1), followed by its application to a range of well-established models (Section 3.2). Finally, we provide an insight into the effectiveness of our method by linking it to perplexity minimization (Section 3.3).

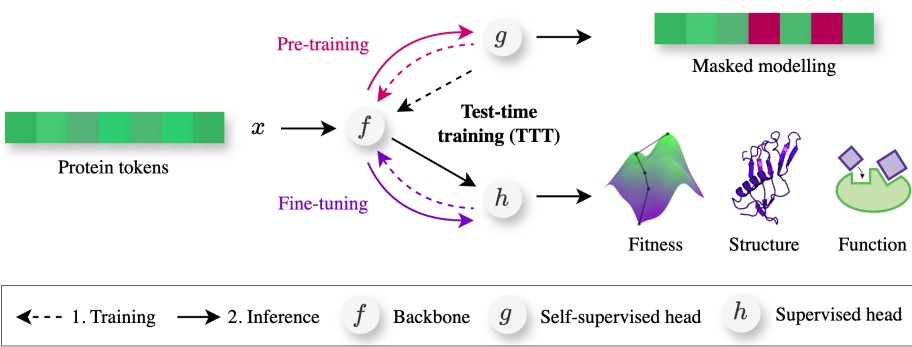

Figure 2: **Overview of our test-time training (TTT) for proteins.** Test-time training for proteins builds on the prevalence of Y-shaped architectures relying on masked modeling (i.e., self-supervised masking-based pre-training of $g \circ f$ followed by supervised fine-tuning of $h \circ f$, sharing the backbone $f$). Given a single test protein $x$, TTT adapts the backbone $f$ to the protein using self-supervised fine-tuning. This adaptation leads to better generalization for the downstream task, such as protein fitness, structure, or function prediction.

### 3.1 SELF-SUPERVISED FINE-TUNING ON TEST PROTEINS

At test time, we assume a Y-shaped model with a backbone $f$ that has been pre-trained via the self-supervised track $g \circ f$, followed by task-specific fine-tuning through the supervised track $h \circ f$. The goal of test-time training (TTT) is to adapt the backbone $f$ to a single test example $x$ before performing test-time inference on a downstream task via the supervised track.

To achieve this, we first fine-tune all layers of the backbone $f$ using the self-supervised track $g \circ f$ on the single example $x$. This step customizes the backbone $f$ to the test sample $x$, and, as demonstrated in Section 4, enhances the generalization of $h \circ f$ without modifying the weights of the task-specific head $h$. Figure 2 illustrates our method. Although the concept of TTT is relatively simple, it involves several important design choices, such as selecting the optimizer and efficiently fine-tuning large backbones, which we describe in the following paragraphs.

**Training objective.** We fine-tune $g \circ f$ on a test sample $x$ via minimizing the masked language modeling objective (Devlin, 2018; Rives et al., 2021):

$$\mathcal{L}(x) = \mathbb{E}_M \left[ \sum_{i \in M} - \log p(x_i | x_{\setminus M}) \right], \tag{1}$$

where $x$ denotes a sequence of protein tokens (typically amino acid types), and $\mathbb{E}_M$ represents the expectation over randomly sampled masking positions $M$. The loss function $\mathcal{L}(x)$ maximizes the log-probabilities $\log p(x_i | x_{\setminus M})$ of the true tokens $x_i$ at the masked positions $i \in M$ in the partially masked sequence $x_{\setminus M}$. Please note that here we focus on bi-directional masked modeling models, which employ random masking, but the method can be straightforwardly extended to models employing autoregressive masking.

In practice, $\mathbb{E}_M$ can follow different distributions, such as sampling a fixed proportion (e.g., 15%) of random amino acid tokens (Lin et al., 2023), or dynamically varying the number of sampled tokens based on another distribution (e.g., a beta distribution) (Hayes et al., 2024). During test-time training, we replicate the masking distribution used during the pre-training. If relevant, we also replicate other pre-training tricks, such as replacing 10% of masked tokens with random tokens and another 10% with the original tokens (Devlin, 2018; Lin et al., 2023; Su et al., 2023) or cropping sequences to random 1024-token fragments (Lin et al., 2023; Su et al., 2023).

**Optimization.** We minimize the loss defined in Equation (1) using stochastic gradient descent (SGD) with zero momentum and zero weight decay (Ruder, 2016). While a more straightforward option might be to use the optimizer state from the final pre-training step, this approach is often impractical because the optimizer parameters are usually not provided with the pre-trained model

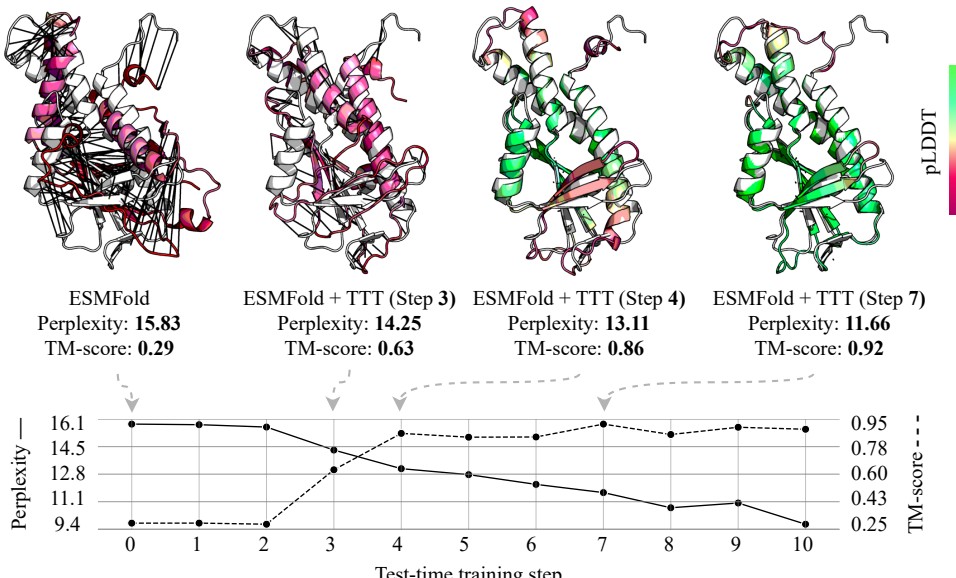

Figure 3: **Test-time training (TTT) improves protein structure prediction by reducing protein sequence perplexity**. ESMFold fails to predict the structure of chain B from PDB entry 7EBL in the CAMEO validation set, as shown at TTT step 0, where the perplexity is high and the TM-score is low. By applying TTT on the single target sequence, the model iteratively improves the structure prediction quality, as demonstrated by the increasing TM-score, associated with reduced perplexity. At step 7, the predicted structure achieves the highest TM-score, as well as the highest predicted confidence metric pLDDT, enabling the selection of this step as the final prediction by ESMFold + TTT.

weights (Hayes et al., 2024; Lin et al., 2023). Moreover, many models are pre-trained using the Adam optimizer (Kingma & Ba, 2015) or its variants (Loshchilov & Hutter, 2019). However, it has been shown that Adam results in less predictable behavior of test-time training (TTT) compared to the SGD optimizer, possibly due to its more exploratory behavior (Gandelsman et al., 2022).

Because each TTT experiment assumes only one test example available, we are not able to halt the training using early stopping on any validation sample. Therefore, for each choice of task-specific $f$ and $h$, we tune the optimal number of TTT steps using the entire validation set beforehand or rely on available performance estimates (e.g., pLDDT in the case of protein structure prediction; Section 4.2) to select the optimal number of optimization steps.

**Fine-tuning large models.** We aim for test-time training to be applicable on the fly, i.e., without the need for any pre-computation and on a single GPU with a minimum computational overhead. Since state-of-the-art models for many protein-oriented tasks are typically large, with up to billions of parameters, our aim presents two key challenges. First, when using pre-trained transformers on a single GPU, even for the forward pass, the batch size is typically limited to only several samples due to the quadratic complexity of the inference (Vaswani, 2017). Second, for the backward pass, even a batch size of one is not always feasible for large models. To address the first challenge, we perform forward and backward passes through a small number of training examples and accumulate gradients to simulate updates with any batch size. We address the second challenge by employing low-rank adaptation (LoRA, Hu et al. (2021)), which in practice enables fine-tuning of any model for which a forward pass on a single sample is feasible, due to a low number of trainable parameters.

## 3.2 INFERENCE ON DOWNSTREAM TASKS

Once the backbone $f$ is adapted to a test protein via self-supervised fine-tuning, it can be used in conjunction with a pre-trained downstream head $h$, as $h \circ f$. The key idea of TTT is not to update the head $h$ during test time, but rather to leverage improved input representations from $f$.

Since Y-shaped architectures are prevalent in protein machine learning, TTT can be straightforwardly applied to numerous tasks in protein research. In this work, we address three primary challenges: protein fitness, structure, and function prediction, applying TTT to corresponding well-established models. For fitness prediction, we apply TTT to ESM2 (Lin et al., 2023) and SaProt (Su et al., 2023); for folding, we apply it to ESMFold (Lin et al., 2023) and ESM3 (Hayes et al., 2024); and for function prediction, we apply TTT to ESM-1v-based (Meier et al., 2021) TerpeneMiner (Samusevich et al., 2024) and ESM-1b-based (Rives et al., 2021) Light attention (Stärk et al., 2021).

In all the models we consider, $f$ is a transformer encoder that takes a protein sequence as input (except for SaProt, which also uses structural tokens), while $g$ is a masked language modeling head (a layer mapping token embeddings to amino acid types). The downstream task heads $h$, however, vary significantly across tasks. For fitness prediction, $h$ outputs a single value for a mutated sequence, measuring how well the protein supports an organism's functioning. Both ESM2 and SaProt perform zero-shot inference using $h \circ f$ via log odds from $g$, with $h$ functioning as a simple adaptation of $g$ without introducing additional parameters. For structure prediction, $h$ is a protein structure decoder: in ESMFold, it is an AlphaFold2-like structure prediction module (Jumper et al., 2021), while in ESM3, it is a VQ-VAE decoder (Razavi et al., 2019). The function predictors are classification models: in TerpeneMiner (Samusevich et al., 2024), $h$ is a random forest that outputs substrate probabilities, and in Light attention (Stärk et al., 2021), $h$ is a light attention module predicting localization class probabilities. Detailed descriptions of the models and their TTT adaptation are provided in Appendix A.

### 3.3 JUSTIFICATION FOR TEST-TIME TRAINING VIA PERPLEXITY MINIMIZATION

While the approach of test-time training has been extensively investigated in computer vision and other domains, the reasons behind its effectiveness remain unclear (Liu et al., 2021; Zhao et al., 2023). Here, we offer a potential justification for the effectiveness of TTT by linking it to perplexity minimization within the context of protein sequence modeling.

Perplexity has traditionally been used in natural language processing to evaluate how well models comprehend test sentences (Brown, 2020; Chelba et al., 2013). Protein language modeling has adopted this metric to assess how effectively models understand amino acid sequences (Hayes et al., 2024; Lin et al., 2023). For bidirectional, random masking language models, which are the focus of this study, we consider the following definition of perplexity [2]:

$$\text{Perplexity}(x) = \exp\left( \frac{1}{|x|} \sum_{i=1}^{|x|} -\log p(x_i | x_{\setminus i}) \right), \quad (2)$$

where $|x|$ is the length of the input protein sequence $x$ and $p(x_i | x_{\setminus i})$ represents the probability that the model correctly predicts the token $x_i$ at position $i$ when it is masked on the input $x_{\setminus i}$. Perplexity ranges from 1 to infinity (the lower the better), providing an intuitive measure of how well a model

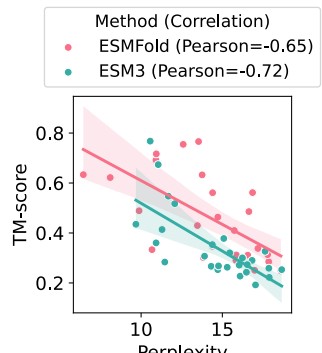

Figure 4: The quality of protein structure prediction, as measured by TM-score, correlates with perplexity of the underlying language model on the challenging targets from the CAMEO validation set. Higher TM-scores are associated with lower perplexity, indicating that better predictions are linked to lower uncertainty in the language model's understanding of the protein sequence.

understands, on average, positions within a given sequence. A perplexity value of 1 indicates that the model perfectly understands the sequence, accurately predicting all the true tokens.

Several studies have shown that lower perplexity on held-out protein sequences (calculated through the self-supervised track $g \circ f$) correlates with better performance on downstream tasks (via the supervised track $h \circ f$), such as predicting protein contacts (Rao et al., 2020), structure (Lin et al., 2023), or fitness (Kantroo et al., 2024). To provide an example, we analyze the correlation between perplexity and structure prediction performance (Figure 4). A strong correlation suggests that

---

[2] Please note that this is an approximation of perplexity, which is computationally intractable for bidirectional models, and is often referred to as pseudo-perplexity (Lin et al., 2023; Salazar et al., 2019).

reducing a model's perplexity on a single test sample $x$ can lead to improved performance on the downstream task (Figure 3; Figure 12).

Since we consider only a single test example $x$, the minimization of the masked language modeling loss $\mathcal{L}(x)$ (Equation (1)) on this example is directly linked to minimizing the perplexity Perplexity$(x)$ (Equation (2)). For instance, in the case of a single masked position (i.e., $|M| = 1$), the loss is equal to the logarithm of perplexity. More generally, it can be shown formally that by minimizing the masked language modeling objective, one learns to approximate the conditional marginals of the language (of proteins), including the leave-one-out probabilities evaluated in perplexity (Hennigen & Kim, 2023). As a result, applying test-time training (TTT) through $g \circ f$ enhances the representation of the test protein in the backbone $f$, leading to improved downstream performance via the fine-tuning track $h \circ f$.

## 4 EXPERIMENTS

Building on the broad applicability of our test-time training (TTT) approach, we apply it to three example downstream tasks in protein machine learning: fitness, structure, and function prediction. The experimental setup and results for each task are presented in the following subsections.

### 4.1 PROTEIN FITNESS PREDICTION

Protein fitness refers to the ability of a protein to efficiently perform its biological function, which is determined by its structure, stability, and interactions with other molecules. Predicting protein fitness allows researchers to understand how mutations affect protein function, aiding in protein engineering (Notin et al., 2024). In this paper, we demonstrate that applying test-time training (TTT) to representative models, such as ESM2 (Lin et al., 2023) and SaProt (Su et al., 2023), enhances their protein fitness prediction capabilities. ESM2 is a protein language model trained on protein sequences, while SaProt is an extension of ESM2 that incorporates 3D information via additional structural tokens encoding structures predicted by AlphaFold2 (Jumper et al., 2021).

**Evaluation Setup.** We evaluate the models using ProteinGym, state-of-the-art benchmark for fitness prediction (Notin et al., 2024), focusing specifically on its well-established zero-shot variant. The zero-shot nature of this benchmark enables us to validate TTT in a simplified setting with a minimalist head $h$, which is complementary to the other tasks described below. Since the zero-shot setup only provides a test set without any data split, we aim to validate TTT on independent data. To achieve this, we create a new fitness prediction dataset mined from MaveDB, a public repository containing datasets from Multiplexed Assays of Variant Effect (MAVEs) (Esposito et al., 2019). The quality of the new dataset is validated by confirming that both ESM2 and SaProt generalize well to the new data, achieving comparable performance (Appendix A).

Given a protein and its variants, fitness prediction models output one real value per variant to estimate fitness. ProteinGym uses Spearman correlation between predicted and experimentally measured fitness values as the main evaluation metric for assessing the capabilities of models to score mutations. The correlation is first calculated for each protein and then aggregated per types of measured fitness: activity, binding, expression, organismal fitness, and stability. The final Spearman correlation metric is obtained by averaging across these five categories. We adopt this metric in our benchmarking.

In our evaluation, we also include other top-performing baselines on the ProteinGym benchmark: TranceptEVE (Notin et al., 2022b) and GEMME (Laine et al., 2019). TranceptEVE combines language model Tranception (Notin et al., 2022a) with the protein-specific variational autoencoder, EVE, capturing the evolutionary information via MSAs (Frazer et al., 2021). GEMME is a statistical method deriving fitness predictions from evolutionary trees.

**Results.** Test-time training (TTT) consistently enhances the protein fitness prediction performance of both ESM2 and SaProt models across varying model scales (35M and 650M parameters) and both datasets, test ProteinGym (Table 1 left) and validation MaveDB (Table 6 in Appendix B.2). Notably, SaProt (650M) + TTT sets a new state-of-the-art on the ProteinGym benchmark, achieving a 40% higher improvement compared to the previous leaderboard update (SaProt (650M) against TranceptEVE L). When examining performance across different phenotype categories, TTT yields

Table 1: **Test-time training (TTT) improves protein fitness prediction.** The right section of the table presents performance averaged across individual proteins and then across different protein phenotypes, as classified in the ProteinGym benchmark (Notin et al., 2024). The middle column shows the final performance, averaged across all five phenotype classes. In total, ProteinGym contains 2.5 million mutations across 217 proteins and TTT is applied to each protein individually. Standard deviations are calculated over 5 random seeds and, for brevity, omitted in the right panel, where the maximum standard deviation does not exceed 0.0004. Methods marked with an asterisk ("*") are the other top-5 methods in ProteinGym, and the metrics are reproduced from the leaderboard (`https://proteingym.org/benchmarks`).

| | Avg. Spearman ↑ | Spearman by phenotype ↑ | | | | |
| | | Activity | Binding | Expression | Organismal Fitness | Stability |
|---|---|---|---|---|---|---|
| ESM2 (35M) (Lin et al., 2023) | 0.3211 | 0.3137 | 0.2907 | 0.3435 | 0.2184 | 0.4392 |
| ESM2 (35M) + TTT (Ours) | **0.3407 ± 0.00014** | **0.3407** | **0.2942** | **0.3550** | **0.2403** | **0.4733** |
| SaProt (35M) (Su et al., 2023) | 0.4062 | 0.3721 | 0.3568 | 0.4390 | 0.2879 | 0.5749 |
| SaProt (35M) + TTT (Ours) | **0.4106 ± 0.00004** | **0.3783** | **0.3569** | **0.4430** | **0.2955** | **0.5795** |
| ESM2 (650M) (Lin et al., 2023) | 0.4139 | 0.4254 | 0.3366 | 0.4151 | 0.3691 | **0.5233** |
| ESM2 (650M) + TTT (Ours) | **0.4153 ± 0.00003** | **0.4323** | **0.3376** | **0.4168** | **0.3702** | 0.5195 |
| TranceptEVE S* (Notin et al., 2022b) | 0.4519 | 0.4750 | 0.3957 | 0.4426 | 0.4491 | 0.4973 |
| GEMME* (Laine et al., 2019) | 0.4547 | 0.4820 | 0.3827 | 0.4382 | 0.4517 | 0.5187 |
| TranceptEVE M* (Notin et al., 2022b) | 0.4548 | 0.4792 | 0.3858 | 0.4525 | 0.4538 | 0.5025 |
| TranceptEVE L* (Notin et al., 2022b) | 0.4559 | 0.4866 | 0.3758 | 0.4574 | 0.4597 | 0.5003 |
| SaProt (650M) (Su et al., 2023) | 0.4569 | 0.4584 | 0.3785 | **0.4884** | 0.3670 | **0.5919** |
| SaProt (650M) + TTT (Ours) | **0.4583 ± 0.00001** | **0.4593** | **0.3790** | 0.4883 | **0.3754** | 0.5896 |

improvements specifically in the categories where the baseline performance is weakest: "Organismal Fitness", "Binding", and "Activity" (Table 1 right). This improvement indicates the ability of TTT to enhance predictions on challenging targets. Additionally, we observe an inverse correlation between the degree of TTT enhancement and the depth of the MSA (i.e., the number of available homologous sequences) available for each test protein, suggesting that TTT primarily improves predictions for proteins with fewer similar sequences available in the training data (Table 5 in Appendix B.1). Interestingly, TTT more effectively enhances the performance of smaller ESM2 and SaProt models compared to their larger variants (Table 1 and Table 6 in Appendix B) and does not require the application of LoRA even for the larger models (Table 4).

### 4.2 PROTEIN STRUCTURE PREDICTION

Protein structure prediction, also known as protein folding, is the task of predicting 3D coordinates of protein atoms given the amino acid sequence. Arguably, one of the most remarkable applications of machine learning in the life sciences has been in protein folding (Jumper et al., 2021; Lin et al., 2023; Abramson et al., 2024), paving the way for numerous advances in the understanding of biology (Yang et al., 2023; Akdel et al., 2022; Barrio-Hernandez et al., 2023). However, even state-of-the-art protein folding methods struggle to generalize to entirely novel proteins (Kryshtafovych et al., 2023). In this work, we focus on the ESMFold (Lin et al., 2023) and ESM3 (Hayes et al., 2024) models, demonstrating how their performance on challenging targets can be boosted by utilizing TTT.

**Evaluation setup.** To evaluate the performance of TTT, we use CAMEO, a standard benchmark for protein folding. We use the validation and test folds from Lin et al. (2023), focusing only on challenging targets by filtering them according to standard measures of prediction confidence based on pLDDT and perplexity (Appendix A.2).

Given a protein sequence, the goal of protein folding is to predict 3D coordinates of the protein atoms. To assess the quality of the predicted protein structures with respect to the ground truth structures, we use two standard metrics: TM-score (Zhang & Skolnick, 2004) and LDDT (Mariani et al., 2013). TM-score measures the quality of the global 3D alignment of the target and predicted protein structures, while LDDT is an alignment-free method based on local distance difference tests.

As baseline methods, we use techniques alternative to TTT for improving the performance of the pre-trained base models. In particular, the ESMFold paper proposes randomly masking 15% of

Table 2: **Test-time training (TTT) improves protein structure prediction.** The metrics are averaged across the 18 challenging targets (TTT is applied to each protein individually) in the CAMEO test set and standard deviations correspond to 5 random seeds. CoT and MP stand for the chain of though and masked prediction baselines.

|  | TM-score ↑ | LDDT ↑ |
|---|---|---|
| ESM3 (Hayes et al., 2024) | 0.3480 ± 0.0057 | 0.3723 ± 0.0055 |
| ESM3 + CoT (Hayes et al., 2024) | 0.3677 ± 0.0088 | 0.3835 ± 0.0024 |
| ESM3 + TTT (Ours) | **0.3954 ± 0.0067** | **0.4214 ± 0.0054** |
| ESMFold (Lin et al., 2023) | 0.4649 | 0.5194 |
| ESMFold + MP (Lin et al., 2023) | 0.4862 ± 0.0043 | 0.5375 ± 0.0070 |
| ESMFold + TTT (Ours) | **0.5047 ± 0.0132** | **0.5478 ± 0.0058** |

amino acids in a protein sequence, allowing for sampling multiple protein structure predictions from the regression ESMFold model (Lin et al., 2023). For each sequence, we sample a number of predictions equal to the total number of TTT steps and refer to this baseline as ESMFold + MP (Masked Prediction). As a baseline for ESM3, we use chain-of-thought iterative decoding, referred to as ESM3 + CoT, proposed in the ESM3 paper (Hayes et al., 2024).

**Results.** Test-time training (TTT) consistently improves the performance of both the ESMFold and ESM3 models, outperforming the masked prediction (ESMFold + MP) and chain-of-thought (ESM3 + CoT) baselines, as shown in Table 2. Of the 18 most challenging CAMEO test proteins, ESMFold and ESM3 significantly improved the prediction of 7 and 6 structures, respectively, while only slightly disrupting the prediction of 2 and 1 structures, respectively (Figure 9 in Appendix B.1). Most notably, TTT enables accurate structure prediction for targets that are poorly predicted with original base models. For instance, Figure 1 presents a strongly improved structure predicted using ESMFold + TTT for the target that was part of the CASP14 competition and shown as an unsuccessful case in the original ESMFold publication (Lin et al. (2023), Fig. 2E). Another example is shown in Figure 3, where TTT refined the structure prediction from a low-quality prediction (TM-score = 0.29) to a nearly perfectly folded protein (TM-score = 0.92). Figure 8 in Appendix B shows that ESMFold + TTT maintains computational efficiency comparable to ESMFold while being orders of magnitude faster than AlphaFold2. Figure 13 in Appendix B additionally demonstrates the robustness of ESM3 + TTT to the choice of hyperparameters.

## 4.3 PROTEIN FUNCTION PREDICTION

Protein function prediction is essential for understanding biological processes and guiding bioengineering but is challenging due to its vague definition and limited data (Yu et al., 2023; Radivojac & et al., 2013; Stärk et al., 2021; Mikhael et al., 2024; Samusevich et al., 2024). While improved structure prediction with TTT (Section 4.2) can already enhance function prediction (Song et al., 2024), we also evaluate TTT directly on two function classification tasks: subcellular localization, predicting protein location within a cell (Stärk et al., 2021), and substrate classification for terpene synthases (TPS), enzymes producing terpenoids, the largest class of natural products (Christianson, 2017; Samusevich et al., 2024). Using TTT with TerpeneMiner (Samusevich et al., 2024) for TPS detection and Light attention (Stärk et al., 2021) for subcellular localization, we achieve consistent performance gains.

**Evaluation setup.** For the terpene substrate classification, we use the largest available dataset of characterized TPS from Samusevich et al. (2024) and repurpose the original cross-validation schema. In the case of protein localization prediction, we use a standard DeepLoc dataset (Almagro Armenteros et al., 2017) as a validation set and setHard from (Stärk et al., 2021) as a test set.

Given a protein, the goal of function prediction is to correctly classify it into one of the predefined functional annotations. We assess the quality of the TPS substrate prediction using standard multi-label classification metrics used in the TerpeneMiner paper (Samusevich et al., 2024): mean average precision (mAP) and area under the receiver operating characteristic curve (AUROC). In the case of protein localization prediction, we similarly use the classification metrics from the original paper (Stärk et al., 2021): accuracy, multi-class Matthews correlation coefficient (MCC), and F1-score.

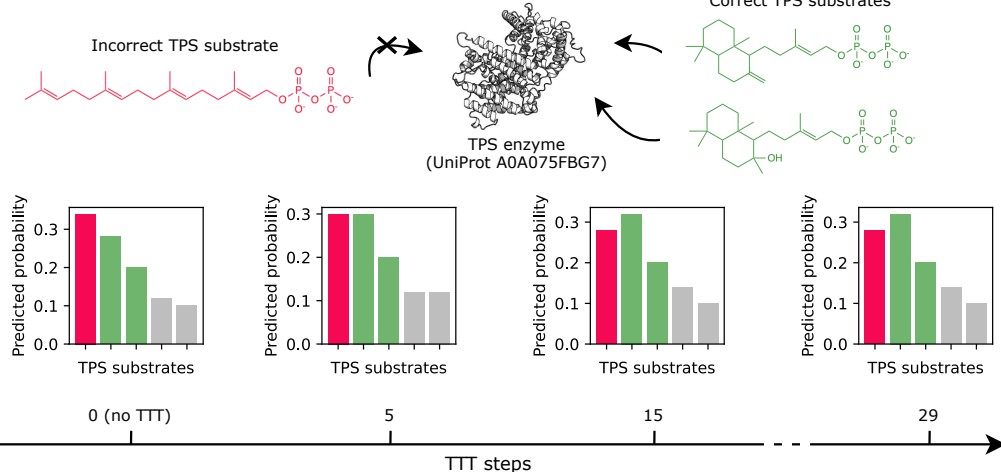

Figure 5: **Test-time training (TTT) enables the correct substrate classification for a terpene synthase (TPS) enzyme.** With progressive test-time training steps of TerpeneMiner + TTT, the probability of the initially misclassified substrate (red) decreases, while the probability of the true substrates (green) increases. The bar plots also display the predicted probabilities for other substrates with non-zero values (grey).

Table 3: **Test-time training (TTT) improves protein function prediction.** For the terpene syntase (TPS) substrate classification task, the metrics are computed on the 512 TPS sequences (TTT is applied to each protein individually) based on the cross-validation schema of the TPS dataset (Samusevich et al., 2024). Subcellular localization prediction performance is reported for 432 protein sequences from the setHard test set (Stärk et al., 2021). The error bars show standard deviations across five random seeds.

| TPS substrate classification | | |
| --- | --- | --- |
| | mAP ↑ | AUROC ↑ |
| TerpeneMiner (Samusevich et al., 2024) | 0.805 | 0.948 |
| TerpeneMiner + TTT (Ours) | **0.811 ± 0.0011** | **0.950 ± 0.0002** |

| Subcellular localization prediction | | | |
| --- | --- | --- | --- |
| | Accuracy ↑ | MCC ↑ | F1-score ↑ |
| Light attention (Stärk et al., 2021) | 0.627 | 0.549 | 0.618 |
| Light attention + TTT (Ours) | **0.634 ± 0.004** | **0.557 ± 0.005** | **0.627 ± 0.004** |

**Results.** TTT improves the performance of the base models on both protein function prediction tasks and across all considered metrics (Table 3). Figure 5 provides a qualitative result, where TTT fine-tuning iteratively refines the prediction of TerpeneMiner toward a correct TPS substrate class.

## 5 DISCUSSION

In this work, we have developed test-time training (TTT) for proteins, enabling per-protein adaptation of machine learning models for enhanced generalization. TTT improves performance across models, their scales, and benchmarks, while primarily enhancing performance on challenging targets. Our results open up the field of self-supervised adaptation for proteins and provide a proof-of-concept for other biology-related domains. While our method demonstrated strong potential, adressing several limitations and researching underexplored directions remain important tasks for future research. Specifically, the success and failure modes of TTT remain unclear, and applying TTT to new tasks requires tuning task-specific hyperparameters. However, our results show that reliable confidence estimates, such as pLDDT, make TTT relatively robust to hyperparameter choices (Figure 13 in Appendix B). Therefore, our future work aims to develop task-agnostic confidence estimates based on protein model representations (Zhang et al., 2024; Rives et al., 2021). Additionally, our findings encourage exploring broader adaptation frameworks for proteins, such as domain adaptation, which leverages both training and test data to address new domains (Ganin & Lempitsky, 2015), and adaptive risk minimization, which employs meta-learning for domain shift adaptation (Zhang et al., 2021).

REPRODUCIBILITY STATEMENT

Our efforts are focused on ensuring that this research is easily reproducible. The proposed test-time training (TTT) method will be released as a Python package, providing easy-to-use wrappers for the models adapted in this paper. Detailed explanations of the application of TTT to individual models and the construction of datasets are included in the appendix. Where applicable, we will also release the source code for dataset generation.

ACKNOWLEDGMENTS

This work was supported by the Ministry of Education, Youth and Sports of the Czech Republic through projects e-INFRA CZ [ID:90254], ELIXIR [LM2023055], CETOCOEN Excellence CZ.02.1.01/0.0/0.0/17_043/0009632, ESFRI RECETOX RI LM2023069. This work was also supported by the European Union (ERC project FRONTIER no. 101097822) and the CETOCOEN EXCELLENCE Teaming project supported from the European Union's Horizon 2020 research and innovation programme under grant agreement No 857560. This work was also supported by the Czech Science Foundation (GA CR) grant 21-11563M and by the European Union's Horizon 2020 research and innovation programme under Marie Skłodowska-Curie grant agreement No. 891397. Views and opinions expressed are however those of the author(s) only and do not necessarily reflect those of the European Union or the European Research Council. Neither the European Union nor the granting authority can be held responsible for them.

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

APPENDIX

In Appendix A, we provide further details on the experimental setup, including comprehensive descriptions of the models, datasets, and metrics used. Next, in Appendix B, we present additional results and their analysis. We discuss the distribution of TTT effects and demonstrate that TTT primarily improves performance on challenging targets. We also explore the impact of hyperparameters by showing the performance on validation sets.

## A EXPERIMENTAL DETAILS

In this section, we describe the experimental details for the three downstream tasks considered in this work: protein fitness prediction (Appendix A.1), protein structure prediction (Appendix A.2), and protein function prediction (Appendix A.3). Each subsection describes the application of test-time training (TTT) to the respective models, along with details on the datasets, metrics, and models. Table 4 additionally summarizes the hyperparameters used for the application of TTT to individual models.

### A.1 PROTEIN FITNESS PREDICTION

#### A.1.1 DATASETS

**ProteinGym.** ProteinGym[3] is the standard benchmark for protein fitness prediction (Notin et al., 2024). The latest, second version of the dataset includes 217 deep mutation scanning experiments (DMSs) across different proteins. We focus on the well-established zero-shot variant of the benchmark and do not experiment with the supervised variant, as it has not yet been fully incorporated into the official codebase at the time of this study. In total, the dataset contains 2.5 mutants with annotated ground-truth fitness. Since ProteinGym does not contain a data split for the zero-shot setup, employed in this work, we use the whole dataset as the test set.

**MaveDB dataset.** To establish a validation set disjoint from ProteinGym (Notin et al., 2024), we mined MaveDB[4] (Esposito et al., 2019). As of August 1, 2024, the database contains 1178 Multiplexed Assays of Variant Effects (MAVEs), where each assay corresponds to a single protein, measuring the experimental fitness of its variants. We applied quality control filters to remove potentially noisy data. Specifically, we ensured that the UniProt identifier (Consortium, 2023) is valid and has a predicted structure available in the AlphaFold DB (Varadi et al., 2022). We also excluded assays with fewer than 100 variants, as well as those where at least one mutation had a wrongly annotated wild type or where most mutations failed during parsing. Additionally, to ensure no overlap between datasets, we removed any assays whose UniProt identifier matched with those in ProteinGym, ensuring that the validation and test sets contain different proteins.

The described methodology resulted in the MaveDB dataset comprising 676 assays (out of 1178 in the entire MaveDB) with experimental fitness annotations. This corresponds to 483 unique protein sequences and 867 thousand mutations in total. The large size of the dataset, despite the comprehensiveness of ProteinGym containing 217 assays, can be attributed to the fact that many assays in MaveDB were released after the ProteinGym construction (Figure 6A). To ensure the quality of the constructed MaveDB dataset, we validated that representative baselines from ProteinGym generalize to the new assays, following a similar distribution of predictions (Figure 6B,C). Finally, for efficiently tuning hyper-parameters for fitness prediction models we sampled 50 random proteins (Figure 6D), corresponding to 83 assays and collectively 134 thousand variants.

#### A.1.2 METRICS

Protein fitness labels are not standardized and can vary across different proteins. Nevertheless, the ranking of mutations for a single protein, as defined by fitness labels, can be used to assess the mutation scoring capabilities of machine learning models. As a result, Spearman correlation is a standard metric for evaluation.

---

[3]https://github.com/OATML-Markslab/ProteinGym
[4]https://www.mavedb.org

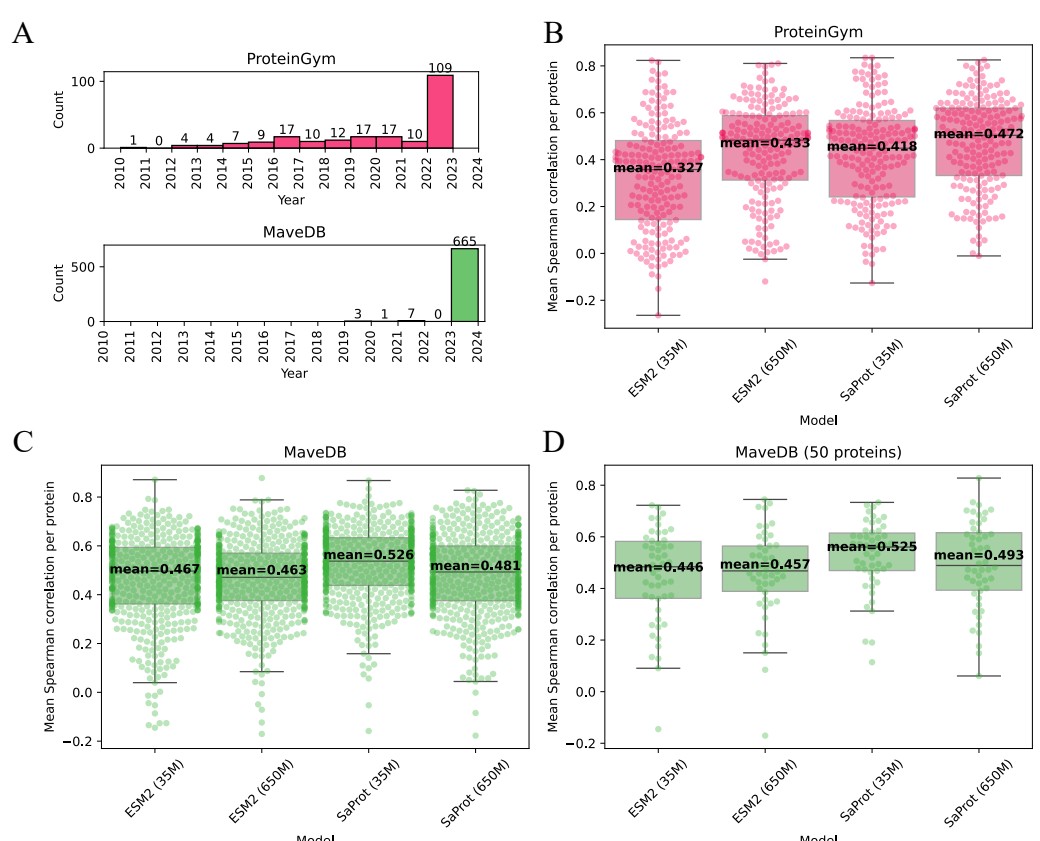

Figure 6: **Comparison of the standard ProteinGym dataset with the MaveDB dataset constructed in this work**. A) MaveDB, mined from Esposito et al. (2019), includes novel assays even after filtering to ensure distinct proteins from the comprehensive ProteinGym dataset. This is largely because most MaveDB assays post-filtering date to 2024, whereas the latest assays in ProteinGym date to 2023. B, C, D) MaveDB is of sufficient quality for model evaluation. Representative baselines, ESM2 and SaProt with both 35 million and 650 million parameters, evaluated on ProteinGym generalize effectively to MaveDB, following a similar distribution of predictions. Panel D illustrates the random subset of 50 proteins used for hyperparameter tuning for fitness prediction. Each point in the plots represents one protein and shows the Spearman correlation averaged across all assays corresponding to the protein (typically one assay per protein). The box plots standardly depict quartiles, medians, and outliers.

**Spearman by phenotype.** When computing Spearman correlations, we follow the evaluation protocol proposed in ProteinGym (Notin et al., 2024). First, for each protein, we compute Spearman correlation scores between the predicted ranks of mutations and their corresponding labels. Then, we average the scores across five categories of assayed phenotypes, measuring the effects of introduced mutations: protein catalytic activity ("Activity"), binding affinity to a target ("Binding"), protein expression levels in a cell ("Expression"), organism growth rate ("Organismal Fitness"), and protein thermostability ("Stability").

**Avg. Spearman.** We refer to the mean score across the five phenotype categories as "Avg. Spearman". We report the "Avg. Spearman" metric as the mean and standard deviation across five random seeds (Table 1, Table 5).

**Spearman by MSA Depth.** Following (Notin et al., 2024), we split the performance by the depth of available multiple sequence alignment (MSA), i.e., the number of homologous sequences available, as provided in ProteinGym: "Low depth", "Medium depth", and "High depth", and report the

Spearman correlation for each subset individually (Table 6). Specifically, the MSA depth categories in ProteinGym are determined using the following thresholds from Hopf et al. (2017): "Low" is defined as $N_{eff}/L < 1$, "Medium" as $1 < N_{eff}/L < 100$, and "High" as $N_{eff}/L > 100$, where $N_{eff}$ represents the normalized number of effective sequences in the MSA, and $L$ is the sequence length covered in the MSA.

### A.1.3 MODELS

**ESM2.** The ESM2 model is a bidirectional, BERT-like (Devlin, 2018) transformer trained on millions of protein sequences using masked modeling (Lin et al., 2023). The goal of protein fitness prediction is to predict the effects of mutations, and protein language models are often adapted to this task using zero-shot transfer via log odds ratio (Notin et al., 2024; Meier et al., 2021). Specifically, for a given single- or multi-point mutation, where certain amino acids $T$ are substituted from $x_i$ to $x_i^m$ for each $i \in T$, the fitness prediction via the log odds ratio is defined as:

$$\sum_{i \in T} \log p(x_i^m | x_{\backslash i}) - \log p(x_i | x_{\backslash i}), \tag{3}$$

where the sum iterates over mutated positions $i \in T$ with $p(x_i^m | x_{\backslash i})$ and $p(x_i | x_{\backslash i})$ denoting the predicted probabilities of the mutated amino acid and the original one (i.e., wild type), respectively. The conditionals $x_{\backslash i}$ indicate that the input sequence to the model has the position $i$ masked. In this setup, the native (unmutated) sequence, where $T = \emptyset$, has a predicted fitness of 0. Mutations with negative values represent favorable mutations, while positive values correspond to disruptive mutations. We follow the ProteinGym benchmark and use this formula (Notin et al., 2024) to evaluate the fitness prediction capabilities of ESM2. We use the implementation of ESM2 from ProteinGym.

**ESM2 + TTT.** ESM2 can be straightforwardly enhanced with test-time training. Specifically, we treat the transformer encoder as the backbone $f$, and the language modeling head, which projects token embeddings to amino acid probabilities, as the pre-training head $g$. The log odds ratio given by Equation (3) serves as the task-specific head $h$, which in this case involves the pre-training head $g$ that predicts log probabilities. Overall, we apply TTT to the pre-trained ESM2 model and, after a pre-defined number of self-supervised fine-tuning steps, score mutations using Equation (3). During TTT we fine-tune all parameters in $g \circ f$ end-to-end except for token and position embeddings.

**SaProt.** We also experiment with the state-of-the-art fitness prediction model, SaProt (Su et al., 2023). SaProt builds off the ESM2 model but incorporates structural information from predicted protein structures. Specifically, SaProt uses the same transformer architecture but expands its vocabulary by combining the 20 standard amino acid tokens with 20 structural tokens from the 3Di vocabulary, increasing the total alphabet size to 400. The 3Di tokens capture the geometry of the protein backbone and are generated using VQ-VAE (Razavi et al., 2019), which projects continuous geometric information into discrete tokens and was trained as part of the Foldseek method (van Kempen et al., 2022).

Since SaProt is also a protein language model, it also uses Equation (3) to score variants. However, please note that SaProt, as implemented in ProteinGym (Notin et al., 2024), uses a slightly different version of the log odds ratio. In SaProt, the conditions in the log probabilities in Equation (3) are replaced with $x_{\backslash T}$ instead of $x_{\backslash i}$, not assuming the independence of substitutions. During TTT, we only mask sequential information and leave the structural part of the tokens unchanged, reflecting the original pre-training setup. We use the implementation of SaProt from ProteinGym[3].

**SaProt + TTT.** Since the architecture of SaProt is based on ESM2, the TTT components $f$, $g$, and $h$ remain the same. It means that test-time training can be applied to the model in the same way as in the case of ESM2 + TTT discussed above.

## A.2 PROTEIN STRUCTURE PREDICTION

### A.2.1 DATASETS

**CAMEO dataset.** To evaluate the capabilities of TTT on protein folding, we employ the CAMEO validation and test sets as described in Lin et al. (2023). Specifically, the validation set was obtained

by querying the CAMEO (Continuous Automated Model Evaluation) web server[5] (Robin et al., 2021) for entries between August 2021 and January 2022, while the CAMEO test set consists of entries from April 1, 2022, to June 25, 2022. Most of the entries in the CAMEO sets are predicted with high accuracy and confidence (Lin et al., 2023). Therefore, we subselected the challenging validation and test sets where TTT is relevant.

Specifically, we applied two criteria: (1) preserving entries with ESMFold pLDDT scores below 70 to filter out high-confidence predictions (Jumper et al., 2021), and (2) selecting entries with ESM2 perplexity scores greater than or equal to 6, ensuring that the predictions are challenging due to poor sequence understanding rather than other factors. Additionally, most structures with perplexity scores below 6 are already associated with high-confidence predictions (Figure S5 in Lin et al. (2023)). After filtering, the resulting challenging validation and test sets consist of 27 (out of 378) and 18 (out of 194) targets, respectively. The vast majority of the remaining structures have accurate ESMFold structure predictions.

### A.2.2 Metrics

To assess the quality of the predicted protein structures with respect to the ground truth structures, we use two standard metrics averaged across the test dataset: TM-score (Zhang & Skolnick, 2004) and LDDT (Mariani et al., 2013).

**TM-score.** The TM-score (Template Modeling score) is a metric used to assess the quality of the global 3D alignment between the predicted and target protein structures. It evaluates the structural similarity by comparing the distance between corresponding residues after superposition. The TM-score ranges from 0 to 1, where higher values indicate better alignment.

**LDDT.** The Local Distance Difference Test (LDDT) is an alignment-free metric used to assess the accuracy of predicted protein structures. Unlike global metrics, LDDT focuses on local structural differences by measuring the deviation in distances between atom pairs in the predicted structure compared to the target structure. It is particularly useful for evaluating the accuracy of local regions, such as secondary structure elements. LDDT scores range from 0 to 100, with higher values indicating better local structural agreement.

### A.2.3 Models

**ESMFold.** The ESMFold architecture comprises two key components: a protein language model, ESM2, which, given a protein sequence, generates embeddings for individual amino acids, and a folding block that, using these embeddings and the sequence, predicts the protein 3D structure along with per-amino-acid confidence scores, known as pLDDT scores. In our experiments, we use the `esmfold_v0` model from the publicly available ESMFold checkpoints[6]. Please note that we use `esmfold_v0` and not `esmfold_v1` to avoid data leakage with respect to the CAMEO test set.

**ESMFold + TTT.** Since ESM2 backbone of ESMFold was pre-trained in a self-supervised masked modeling regime, the application of TTT to ESMFold is straightforward. We treat ESM2 as the backbone $f$, the language modeling head predicting amino acid classes from their embeddings as the self-supervised head $g$, and the folding trunk along with the structure modules as the downstream task head $h$. After each TTT step, we run $h \circ f$ to compute the pLDDT scores, which allows us to estimate the optimal number of TTT steps for each protein based on the highest pLDDT score.

Since the backbone $f$ is given by the ESM2 model containing 3 billion parameters, we apply LoRA (Hu et al., 2021) to all matrices involved in self-attention. This enables fine-tuning ESMFold + TTT on a single GPU.

**ESMFold + ME.** Since ESMFold is a regression model, it only predicts one solution and does not have a straightforward mechanism of sampling multiple structure predictions. Nevertheless, the authors of ESMFold propose a way to sample multiple candidates (Section A.3.2 in Lin et al. (2023)).

---

[5]https://www.cameo3d.org/modeling

[6]https://github.com/facebookresearch/esm/blob/main/esm/esmfold/v1/pretrained.py

To sample more solutions, the masking prediction (ME) method randomly masks 15% (same ratio as during masked language modeling pre-training) of the amino acids embeddings before passing them to the structure prediction block. Selecting the solution with the highest pLDDT may lead to improved predicted structure. Since sampling multiple solutions with ESMFold + ME and selecting the best one via pLDDT is analogous to ESMFold + TTT, we employ the former as a baseline, running the method for the same number of step.

**ESM3.** Unlike ESMFold, ESM3 is a fully multiple-track, BERT-like model (Devlin, 2018), pre-trained to unmask both protein sequence and structure tokens simultaneously (along with the function tokens). The structure tokens in ESM3 are generated via a separately pre-trained VQ-VAE (Razavi et al., 2019) operating on the protein geometry. In our experiments, we use the smallest, publicly available version of the ESM3 model (`ESM3_sm_open_v0`)[7].

**ESM3 + TTT.** We treat the transformer encoder of ESM3 as $f$, the language modeling head decoding amino acid classes as $g$, and the VQ-VAE decoder, which maps structure tokens to the 3D protein structure, as $h$. During the TTT steps, we train the model to unmask a protein sequence while keeping the structural track fully padded. During the inference, we provide the model with a protein sequence and run it to unmask the structural tokens, which are subsequently decoded with the VQ-VAE decoder. After each TTT step, we run $h \circ f$ to compute the pLDDT scores, which allows us to estimate the optimal number of TTT steps for each protein based on the highest pLDDT score. We choose the optimal hyperparameters by maximizing the difference in TM-score after and before applying TTT across the validation dataset.

Despite the fact that the model contains 1.4 billion parameters, even without using LoRA, ESM3 + TTT can be fine-tuned on a single NVIDIA A100 GPU. Therefore, we do not employ LoRA for fine-tuning ESM3.

**ESM3 + CoT.** To improve the generalization and protein-specific performance of ESM3, the original ESM3 paper employs a chain of thought (CoT) procedure. The procedure unfolds in $n$ steps as follows. At each step, $1/n$ of the masked tokens with the lowest entropy after softmax on logits are unmasked. Then, the partially unmasked sequence is fed back into the model, and the process repeats until the entire sequence is unmasked. In our experiments, we set $n = 8$, which is the default value provided in the official GitHub repository.

### A.3 PROTEIN FUNCTION PREDICTION

#### A.3.1 DATASETS

**TPS dataset.** For the evaluation of terpene substrate classification, we use the largest available dataset of characterized TPS enzymes from Samusevich et al. (2024) and repurpose the original 5-fold cross-validation schema. We focus on the most challenging TPS sequences, defined as those predicted by the TPS detector, proposed by the dataset authors, with confidence scores below 0.8. This filtering results in 104, 98, 113, 100, 97 examples in the individual folds.

**setHard.** For the test evaluation of subcellular location prediction, we use the setHard dataset constructed by Stärk et al. (2021). The dataset was redundancy-reduced, both within itself and relative to all proteins in DeepLoc (Almagro Armenteros et al. (2017); next paragraph), a standard dataset used for training and validating machine learning models. The setHard dataset contains 490 protein sequences, each annotated with one of ten subcellular location classes, such as "Cytoplasm" or "Nucleus". Since we use ESM-1b (Rives et al., 2021) in our experiments with the dataset, we further filter the data to 432 sequences that do not exceed a length of 1022 amino acids. This step, consistent with Stärk et al. (2021), ensures that ESM-1b can generate embeddings for all proteins.

**DeepLoc.** For hyperparameter tuning in the subcellular location prediction task, we use the test set from the DeepLoc dataset (Almagro Armenteros et al., 2017). Similar to setHard, DeepLoc assigns labels from one of ten subcellular location classes. The dataset contains 2768 proteins, which we further filter to 2457 sequences that do not exceed a length of 1022 amino acids, ensuring

---

[7]https://github.com/evolutionaryscale/esm

compatibility with the embedding capabilities of ESM-1b. Since setHard was constructed to be independent of DeepLoc, setHard provides a leakage-free source of data for validation.

### A.3.2 METRICS

**mAP, AUROC.** The TPS substrate prediction problem is a 12-class multi-label classification task over possible TPS substrates. Therefore, we assess the quality of the predictions using standard multi-label classification metrics such as mean average precision (mAP) and area under the receiver operating characteristic curve (AUROC) averaged across individual classes. These metrics were used in the original TerpeneMiner paper (Samusevich et al., 2024). We report the performance by averaging the metric values concatenated across all validation folds from the 5-fold cross-validation schema.

**Accuracy, MCC, F1-score.** To evaluate the performance of subcellular location prediction methods, we use standard classification metrics as employed in Stärk et al. (2021). Accuracy standardly measures the ratio of correctly classified proteins, while Matthew's correlation coefficient for multiple classes (MCC) serves as an alternative to the Pearson correlation coefficient for classification tasks (Gorodkin, 2004). The F1-score, the harmonic mean of precision and recall, evaluates performance from a retrieval perspective, balancing the trade-off between false positives and false negatives.

### A.3.3 MODELS

**TerpeneMiner.** TerpeneMiner is a state-of-the-art method for the classification of terpene synthase (TPS) substrates (Samusevich et al., 2024). The model consists of two parallel tracks. Given a protein sequence, TerpeneMiner first computes its ESM-1v embedding (Meier et al., 2021) and a vector of similarities to the functional domains of proteins from the training dataset, based on unsupervised domain segmentation of AlphaFold2-predicted structures (Jumper et al., 2021). The ESM-1v embedding and the similarity vector are then concatenated and processed by a separately trained random forest, which predicts TPS substrate class probabilities.

In our experiments, we use the "PLM only" version of the model, which leverages only ESM-1v embeddings (PLM stands for protein language model). This version exhibits a minor performance decrease compared to the full model but exactly follows a Y-shaped architecture, allowing us to validate the effectiveness of test-time training for predicting TPS substrates. We use the implementation of TerpeneMiner available at the official GitHub page [8].

**TerpeneMiner + TTT.** When applying TTT to TerpeneMiner, we treat the frozen ESM-1v model as a backbone $f$, its language modeling head as a self-supervised head $g$, and the random forest classifying TPS substrates as a downstream supervised head $h$.

**Light Attention.** We use Light attention (Stärk et al., 2021) as a representative baseline for subcellular location prediction. Light attention leverages protein embeddings from a language model, which in our case is ESM-1b (Rives et al., 2021). The model processes per-residue embeddings via a softmax-weighted aggregation mechanism, referred to as light attention, which operates with linear complexity relative to sequence length and enables richer aggregation of per-residue information, as opposed to standard mean pooling. We re-train the model using ESM-1b embeddings on the DeepLoc dataset (Almagro Armenteros et al., 2017) using the code from the official GitHub page [9].

**Light attention + TTT.** When applying TTT to Light attention, we treat the frozen ESM-1b as the backbone $f$, the language modeling head of ESM-1b as the self-supervised head $g$, and the Light attention block as the fine-tuning head $h$.

## B EXTENDED RESULTS

In this section, we provide additional results on test sets (Appendix B.1) and discuss validation performance (Appendix B.2).

---

[8] https://github.com/pluskal-lab/TerpeneMiner
[9] https://github.com/HannesStark/protein-localization

Table 4: **Hyperparameters used for adapting TTT to individual models.** The optimal hyperparameters were estimated using validation datasets corresponding to each of the considered tasks: *Fitness prediction*, *Structure prediction*, and *Function prediction*. Comma-separated lists show the values used for hyperparameter grid search, while the final values selected for computing the test results are highlighted in **bold**. Low-rank adaptation (LoRA) was only used with ESMFold, containing 3 billion parameters in the ESM2 backbone. Please note that we did not tune the number of TTT steps, as adjusting the learning rate and batch size effectively controls the expected performance under the fixed number of steps, as shown in Figure 12. Therefore, we used 30 steps in all our experiments. The only exception was ESM3 + TTT, where the number of steps was set to 50 during initial experiments with different models/tasks conducted in parallel before standardizing the number of steps to 30.

|  | Learning rate | Batch size | Grad. acc. steps | TTT steps | LoRA rank $r$ | LoRa $\alpha$ |
|---|---|---|---|---|---|---|
| *Fitness prediction* | | | | | | |
| ESM2 (35M) + TTT | 4e-5, **4e-4**, 4e-3 | **4** | 4, 8, **16**, 32, 64 | **30** | -, 4, 8, 32 | -, 8, 16, 32 |
| ESM2 (650M) + TTT | **4e-5**, 4e-4, 4e-3 | **4** | 4, 8, **16**, 32 | **30** | -, 4, 8, 32 | -, 8, 16, 32 |
| SaProt (35M) + TTT | 4e-5, **4e-4**, 4e-3 | **4** | 4, **8**, 16, 32 | **30** | - | - |
| SaProt (650M) + TTT | **4e-5**, 4e-4, 4e-3 | 2, 4 | 4, 8, **16**, 32 | **30** | - | - |
| *Structure prediction* | | | | | | |
| ESMFold + TTT | **4e-4** | **4** | **4**, 8, 32, 64 | **30 (max pLDDT)** | 4, **8**, 32 | 8, 16, **32** |
| ESM3 + TTT | 1e-4, 4e-4, **1e-3** | **2** | **1**, 4, 16 | **50 (max pLDDT)** | - | - |
| *Function prediction* | | | | | | |
| TerpeneMiner + TTT | **4e-4**, 1e-3 | **2** | **2**, 4, 8 | **30** | - | - |
| Light attention + TTT | 4e-4, 1e-3, **3e-3** | **2** | **2**, 4 | **30** | - | - |

## B.1 DETAILED TEST PERFORMANCE

In this section, we provide details on the test performance. Specifically, Table 5 shows that test-time training (TTT) primarily enhances performance on challenging targets, characterized by a low number of similar proteins in sequence databases, as measured by MSA depth. Additionally, we provide an example illustrating how TTT substantially improves the correlation between ESM2-predicted fitness and ground-truth stability by better identifying disruptive mutations in the protein core (Figure 7).

Next, Figure 9 shows the distribution of TTT effects: in many cases, TTT has minimal impact on performance; often, it leads to substantial improvements; and in rare cases TTT results in a decrease in performance. This positions TTT as a method for enhancing prediction accuracy, while a comprehensive analysis of its failure modes remains an important direction for future research. While we demonstrate these effects using a protein folding example, we observe a similar distribution of TTT impact across the tasks.

We also observe that the overall trend of TTT generally leads to improved performance, with robust consistency across random seeds. However, the progression of the performance curve can be rugged, particularly in classification tasks, where substantial changes in the underlying representations are required to shift the top-predicted class in the discrete probability distribution (Figure 11).

## B.2 VALIDATION PERFORMANCE

This section discusses the performance of test-time training (TTT) on validation data. Table 6 illustrates the validation performance of all tested methods for fitness prediction on our newly constructed MaveDB dataset. TTT enhances the performance of all the methods.

The primary focus of the section is hyperparameter tuning. Table 4 provides the grid of hyperparameters explored for each model and its size. Figure 12 demonstrates the trend of hyperparameter tuning with optimal hyperparameter combination balancing underfitting and overfitting to a single test protein. While most hyperparameter configurations lead to overall improvements when using TTT, poorly chosen hyperparameters can have detrimental effects due to rapid overfitting. However, with a reliable predicted confidence measure, such as pLDDT, the appropriate TTT step can be selected to mitigate overfitting. Figure 13 demonstrates that when using ESM3 + TTT with pLDDT-based step selection for protein folding, all hyperparameter configurations result in improved performance compared to the base ESM3 model.

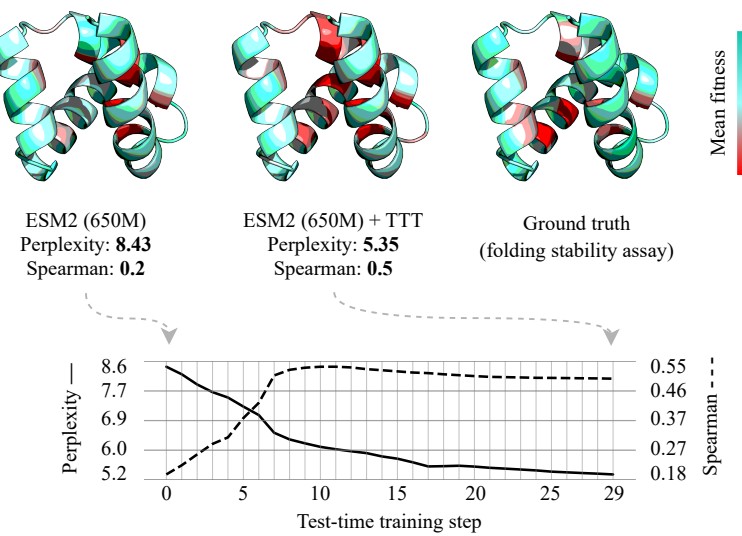

Figure 7: **Example of test-time training (TTT) applied to fitness prediction.** Fitness predictions from ESM2 (650M) show poor correlation with experimental fitness values in the ProteinGym test set measured by the stability assay "UBR5_HUMAN_Tsuboyama_2023_1I2T" (Tsuboyama et al., 2023) (left). ESM2 + TTT achieves significantly higher correlation, likely due to improved detection of disruptive mutations in the protein core that impact protein stability (middle). The ground-truth fitness data aligns with the TTT-enhanced model, showing that residues crucial for stability (i.e., having negative mean fitness) are concentrated in the protein core (right). Residue colors represent the mean fitness upon all single-point substitutions (with the exception of several missing mutations in the ground-truth data), with red indicating residues where mutations have detrimental effects on average.

Table 5: **Test-time training (TTT) performance on ProteinGym depending on MSA depth.** MSA depth reflects the number of available proteins similar to the target protein and, when using large protein language models, can be interpreted as a measure of the representation of similar proteins in the training data (Appendix A.1.2). TTT primarily improves performance on difficult targets, with low MSA depth. Standard deviations are calculated over 5 random seeds but are omitted in the right panel for brevity, where the maximum standard deviation does not exceed 0.0004.

| | Avg. Spearman ↑ | Spearman by MSA depth ↑ | | |
| --- | --- | --- | --- | --- |
| | | Low depth | Medium depth | High depth |
| ESM2 (35M) (Lin et al., 2023) | 0.3211 | 0.2394 | 0.2707 | 0.451 |
| ESM2 (35M) + TTT (Ours) | **0.3407 ± 0.00014** | **0.2445** | **0.3144** | **0.4598** |
| SaProt (35M) (Su et al., 2023) | 0.4062 | 0.3234 | 0.3921 | 0.5057 |
| SaProt (35M) + TTT (Ours) | **0.4106 ± 0.00004** | **0.3253** | **0.3972** | **0.5091** |
| ESM2 (650M) (Lin et al., 2023) | 0.4139 | 0.3346 | 0.4063 | **0.5153** |
| ESM2 (650M) + TTT (Ours) | **0.4153 ± 0.00003** | **0.3363** | **0.4126** | 0.5075 |
| SaProt (650M) (Su et al., 2023) | 0.4569 | 0.3947 | **0.4502** | **0.5448** |
| SaProt (650M) + TTT (Ours) | **0.4583 ± 0.00001** | **0.3954** | 0.4501 | 0.5439 |

Table 6: **Performance of test-time training (TTT) on the MaveDB dataset.** In this work, we use our newly constructed MaveDB benchmark as a validation fold for tuning the hyper-parameters of TTT for fitness prediction. For computational efficiency, we only select a subset of 50 proteins (Appendix A.1.1) and do not run TTT across multiple random seeds to estimate standard deviations. The performance shown was calculated by first aggregating correlations per assay, and then per protein (some assays correspond to the same protein).

| | Avg. Spearman ↑ |
| --- | --- |
| ESM2 (35M) (Lin et al., 2023) | 0.4458 |
| ESM2 (35M) + TTT (Ours) | **0.4593** |
| ESM2 (650M) (Lin et al., 2023) | 0.4568 |
| ESM2 (650M) + TTT (Ours) | **0.4604** |
| SaProt (650M) (Su et al., 2023) | **0.4926** |
| SaProt (650M) + TTT (Ours) | **0.4926** |
| SaProt (35M) (Su et al., 2023) | 0.5251 |
| SaProt (35M) + TTT (Ours) | **0.5271** |

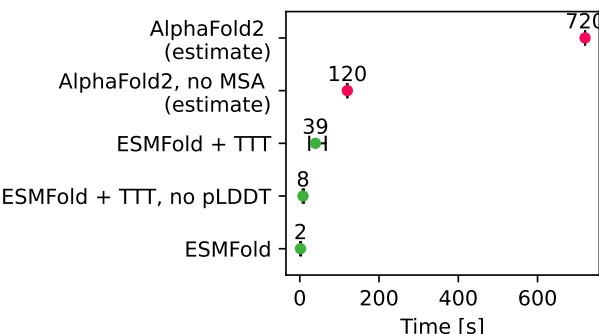

Figure 8: **Running time of ESMFold + TTT.** For ESMFold and its variants, the median and interquartile ranges of running times on the CAMEO test set are shown using a single NVIDIA A100 GPU. For AlphaFold2, we use estimates from Lin et al. (2023). Specifically, a forward pass through AlphaFold2 is approximately 60 times more computationally expensive than ESMFold (e.g., `AlphaFold2, no MSA`: $2 \times 60 = 120$ seconds), with additional MSA construction taking at least 10 minutes using standard pipelines (`AlphaFold2`: $2 \times 60 + 10 \times 60 = 720$ seconds). ESMFold + TTT (30 steps) involves test-time training parameter updates with LoRA, along with forward passes at each TTT step to estimate pLDDT and select the structure with the highest predicted confidence. Disabling pLDDT significantly reduces computational overhead (`ESMFold + TTT, no pLDDT` compared to `ESMFold + TTT`), but may require careful parameter tuning (Appendix B.2). Overall, ESMFold + TTT maintains the speed advantage of ESMFold, and is significantly faster than AlphaFold2.

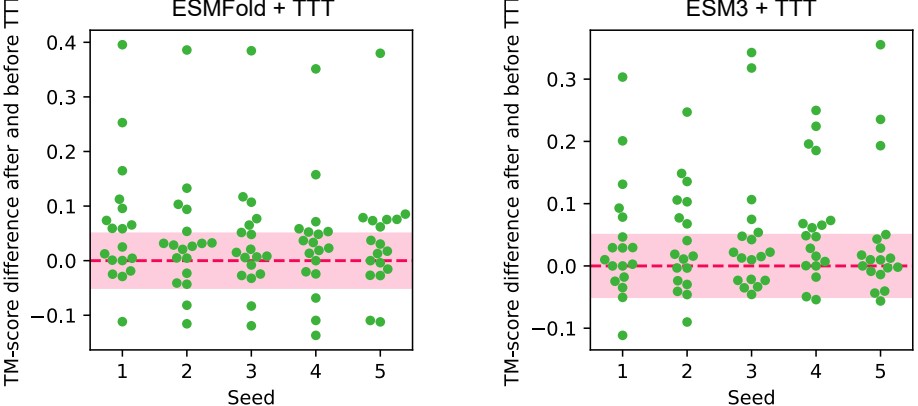

Figure 9: **Per-protein performance of ESMFold + TTT and ESM3 + TTT on the CAMEO test set.** The y-axis shows the change in TM-score after applying test-time training (TTT), with higher values indicating improvement. The x-axis represents performance across five random seeds. The red dashed line marks no change in TM-score (TM-score difference $= 0$), and the pink band represents minor changes in TM-score ($-0.05 <$ TM-score difference $< 0.05$), which we do not consider significant. Each point in the swarm plot corresponds to a single protein from the CAMEO test set. On average, applying TTT to ESMFold improves the structure predictions for 7 out of 18 proteins, with 2 showing degradation. The rest of the proteins are not significantly affected. Similarly, applying TTT to ESM3 results in 6 improvements out of 18 proteins, with 1 case of degradation.

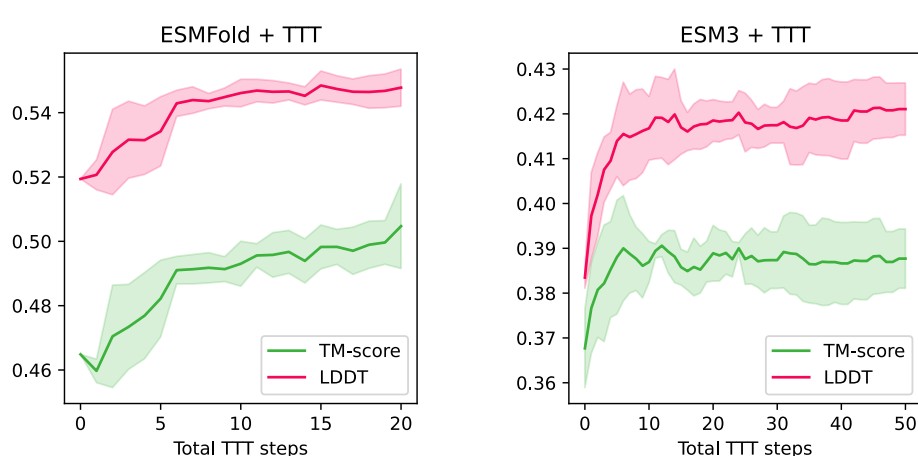

Figure 10: **Test performance of ESMFold + TTT and ESM3 + TTT on the CAMEO test set depending on the total number of TTT steps.** The x-axis shows the averaged performance across all test proteins, with error bars representing the standard deviation across five random seeds. The y-axis metrics correspond to the structure with the highest pLDDT score up to the given step. While an increased number of TTT steps generally enhances performance, only a few TTT steps (e.g., five) may suffice to achieve significant performance improvement.

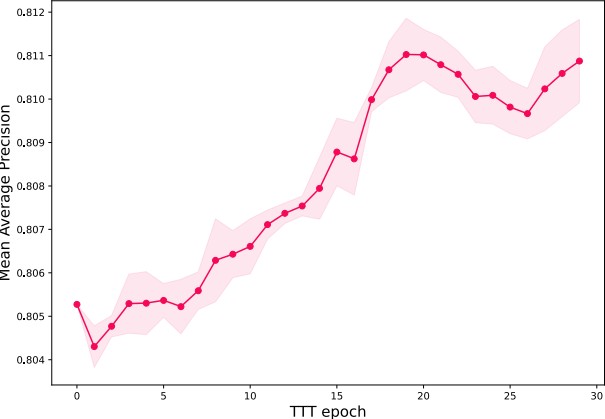

Figure 11: **Test performance of TerpeneMiner + TTT across fine-tuning steps.** The performance is averaged across all 512 proteins in the dataset, with error bars representing the standard deviation across 5 random seeds.

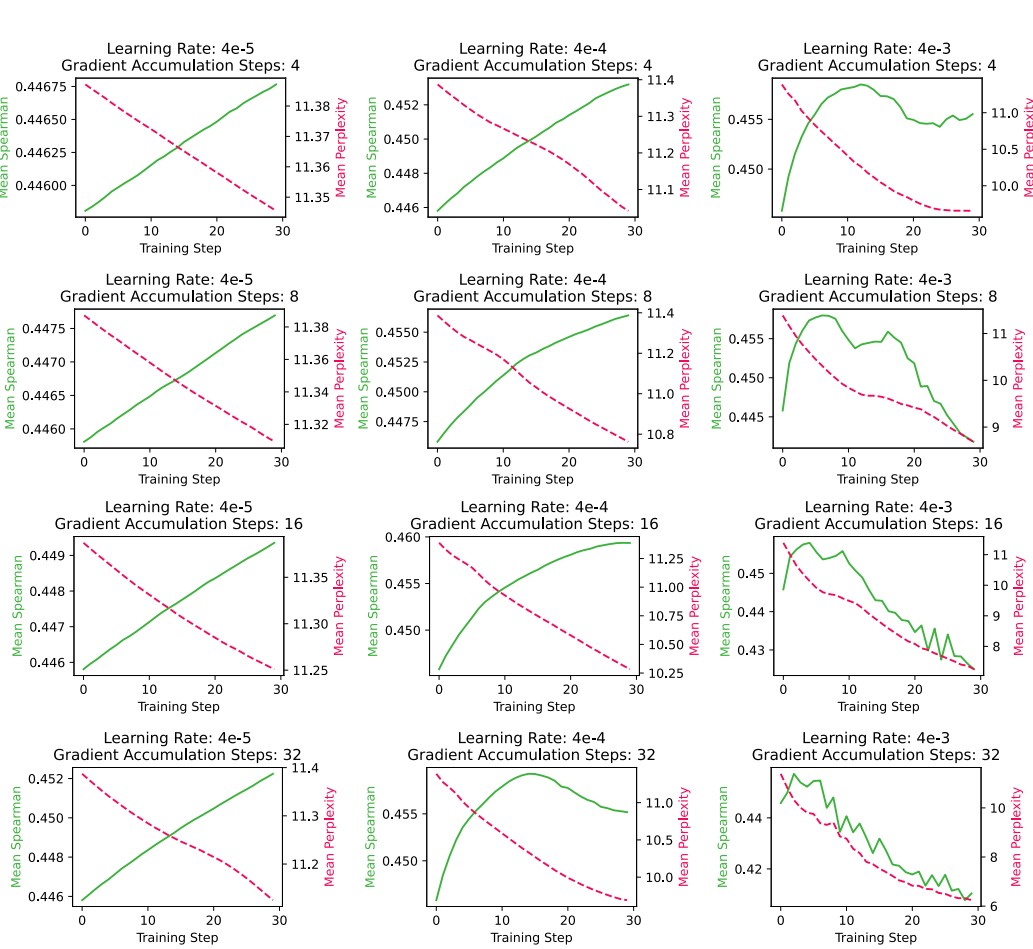

Figure 12: **Dependence on hyperparameters in test-time training for fitness prediction.** Each plot shows the progression of Spearman correlation (green) increasing alongside a decrease in perplexity (pink) for each TTT step, averaged across all assays in the MaveDB validation dataset. The model used is ESM2 (35M) + TTT, and the grid displays the combinations of different numbers of gradient accumulation steps (i.e., effective batch sizes; shown in rows, increasing from top to bottom) and learning rates (columns, increasing from left to right). As the learning rate increases and the number of gradient accumulation steps grows, the model reaches peak performance more quickly but begins to overfit to a test protein. The optimal hyperparameter combination (learning rate = 4e-4, gradient accumulation steps = 16) lies near the center of the grid, balancing between underfitting and overfitting to a test protein. Notably, the figure demonstrates that, although TTT involves three main hyperparameters (batch size, learning rate, and the number of TTT steps), there are effectively only two degrees of freedom controlling the performance of the model. In other words, by keeping the number of steps constant (e.g., 30), the expected performance can be controlled by adjusting the learning rate and the batch size.

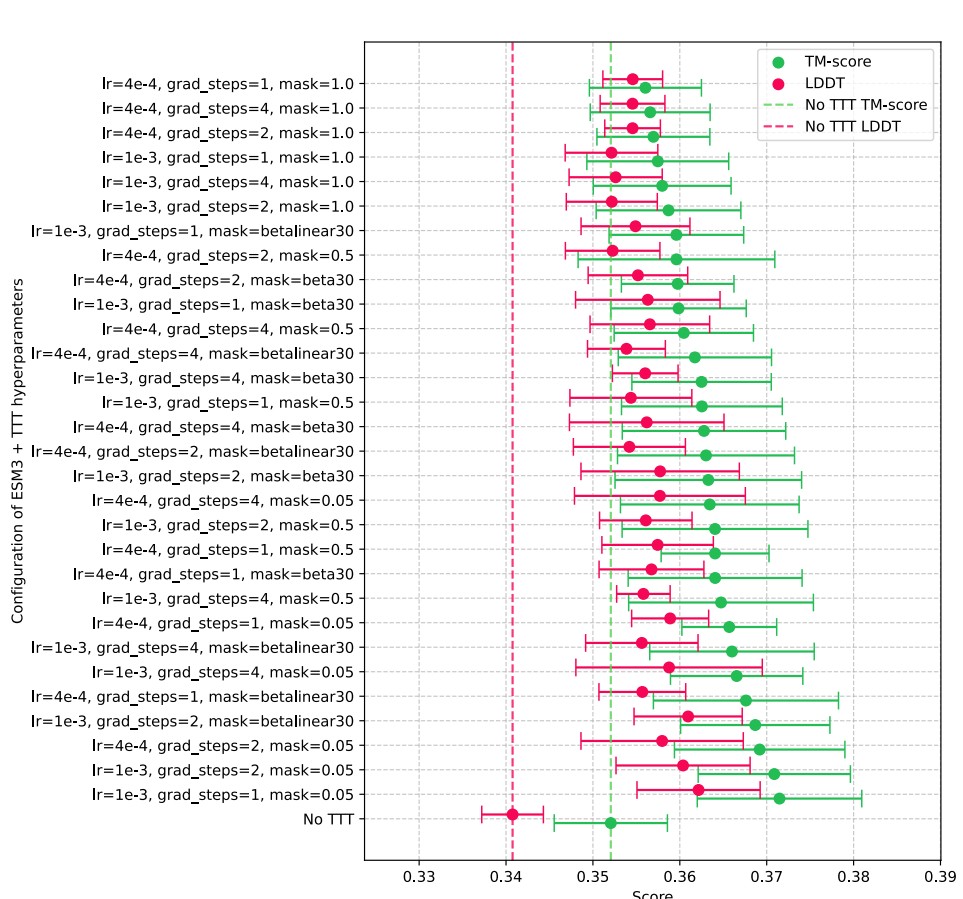

Figure 13: **Hyperparameter search for protein structure prediction with ESM3 + TTT.** We conducted a comprehensive grid search based on three key hyperparameters: learning rate (denoted as "lr"), number of gradient accumulation steps (denoted as "grad_steps"; with the batch size of two), and masking strategy (denoted as "mask"). We explored two learning rates, 4e-4 and 1e-3, three gradient accumulation step values of 1, 4, and 16, and five different masking strategies: uniform sampling of 0.05, 0.5, and 1.0 fractions of amino acids, as well as the beta30 and betalinear30 distributions proposed in the ESM3 paper (Hayes et al., 2024). Each row in the table presents the mean TM-score and LDDT metrics with standard deviations across five random seeds on the CAMEO validation fold. The last row, denoted as "No TTT", shows the performance of ESM3 without TTT. The results indicate that ESM3 + TTT is robust to the choice of hyperparameters and consistently outperforms the base model across all configurations. We selected the configuration from the last row (excluding "No TTT") to compute the results on the test fold. For the hyperparameter search, we used 30 TTT steps instead of 50 to reduce computation time.

