# OpenReview forum: "Training on test proteins improves fitness, structure, and function prediction"
_ICLR.cc/2025/Conference — Submitted to ICLR 2025_

### Official Review · Reviewer_Rmaa · 2024-10-28

**Soundness:** 2
**Presentation:** 2
**Contribution:** 2
**Rating:** 3
**Confidence:** 4

**Summary:**

The authors propose a method to perform test-time adaption of protein masked language models (MLM) for improved supervised prediction of protein properties. The method partially minimizes the standard MLM loss on a single test example before a property prediction is made from the supervised head. The authors explain that the success of this method may be due to the fact that it reduces the perplexity of masked token predictions for the target protein and lower perplexity has been shown to correlate with improved property prediction. The authors test their method on three tasks: (i) comprehensive zero-shot prediction benchmarking in ProteinGym, (ii) structure prediction with ESMFold and (iii) two supervised classification tasks (substrate prediction for a particular class of enzymes and protein localization prediction).

**Strengths:**

* The paper is clearly written and organized.
* The method is described clearly and is well-motivated by existing literature.
* The figures are nicely designed and clearly convey the intended information.
* The evaluations covers a broad range of tasks in fitness prediction.

**Weaknesses:**

1. The method is motivated by models with "Y-shaped" architectures in which there is both an MLM head and a fitness prediction head. However, the authors evaluate on the zero-shot task in ProteinGym, where models use masked token probabilities as proxies for fitness predictions and therefore do not use a Y-shaped architecture. This is confusing, since ProteinGym also contains fine-tuning tasks that require the Y-shaped architectures that motivate their method. The paper would be improved by evaluating on the fine-tuning tasks in ProteinGym or by more clearly explaining why the zero-shot task is more appropriate.
2. In "Background and Related Work", the authors cite a number of papers that performed model adaptation on either protein MSAs or otherwise related proteins to the target protein. These methods are then dismissed because "constructing MSAs is time-consuming, and similar proteins may not be available for many targets." These arguments are situational and are not strong enough reasons to avoid direct comparisons with the cited methods. For example, one may be making predictions for many closely related target proteins, in which case constructing a single MSA and using that to adapt the model for all target proteins (as in TranceptEVE) may be faster than applying TTT to each target. The paper would be strengthened by including a more in-depth discussion about the distinction between existing methods and the authors' proposed method, a discussion about the situations in which each type of method is relevant, and a quantitative experiment demonstrating improved performance in a task between the proposed method and the existing methods.
    * A reasonable comparison is to compare the difference in performance between TranceptEVE and Tranception to the difference in performance between methods w/o TTT to those using TTT (e..g SaProt (650M) and SaProt (650M)+TTT). On avg spearman, the difference between TranceptEVE and Tranception is about 0.02, while the difference between methods with and without TTT is usually about an order magnitude smaller (0.0014 for SAProt (650M) and SAProt (650M)+TTT). This is not a perfect comparison, but it does suggest that the MSA-based adaption of TranceptEVE can have advantages over TTT, which warrants further investigation about the distinction and relative advantages of the methods.
3. The result on improving structure prediction in ESM3 and ESMFold is interesting, but of limited practical relevance since AlphaFold2/3 are far superior structure predictors. The paper would be strengthened by either demonstrating an application of TTT to AlpahFold or an explanation of why this is not possible.
4. Less important: the authors claim in section 3.3 to offer an explanation for the effectiveness of test time adaptation by appealing to the correlation between perplexity minimization and model performance. This is somewhat circular, since the proposed method directly lowers the perplexity of the target protein. Therefore, the positive results provide mre evidence that perplexity minimization is important, but don't provide an explanation for why the method works. I recommend framing this as a justification for the method rather than an explanation for why it works.

**Minor**:
* Line 113-114: " For example, in AlphaFold2, most of the loss function weight is put onto masked modeling of multiple sequence alignment". I believe the authors are referring to the fact that the MSA loss has a weight of 2 while the FAPE loss has a weight of 0.5. This may represent a difference in scales in the different losses rather the relative importance of the losses to training. This line should be removed.
* Line 189 typo: $h \circ g$ should be $h \circ f$
* Line 375 typo: both instances of "TraceptEVE" should say "TranceptEVE"

**Questions:**

* Why did the authors choose to evaluate on the zero-shot tasks in ProteinGym, rather than the fine-tuning tasks? Is there literature precedent for using test time adaption to improve zero-shot prediction?
* What is the rough time scaling of TTT? How does it compare to the scaling of MSA generation?
* Can the authors demonstrate a scenario in which TTT clearly outperforms the MSA-based adaptation of TranceptEVE?
* Is it possible to apply TTT to AlphaFold or OpenFold?

---

> ### Author Response · Authors · 2024-11-22
>
> > **W1.** The method is motivated by models with "Y-shaped" architectures in which there is both an MLM head and a fitness prediction head. However, the authors evaluate on the zero-shot task in ProteinGym, where models use masked token probabilities as proxies for fitness predictions and therefore do not use a Y-shaped architecture. This is confusing, since ProteinGym also contains fine-tuning tasks that require the Y-shaped architectures that motivate their method. The paper would be improved by evaluating on the fine-tuning tasks in ProteinGym or by more clearly explaining why the zero-shot task is more appropriate.
>
> **We have clarified that the zero-shot setup also follows the Y-shaped architecture**. Please note that in the zero-shot setup, the Y-shaped architecture is given by the language model backbone $f$ (e.g., ESM2 Transformer encoder), the pre-training head $g$ given by the MLM head, and the downstream-task head $h$ given by the standardly used log odds ratio. This may not be obvious to realize since, in this case, $h$ contains $g$. We have clarified this aspect in Section 3.2.
>
> **We have clarified the choice of the zero-shot ProteinGym benchmark as the best established benchmark for fitness prediction.** The goal of our paper is to demonstrate the potential of TTT across well-established problems and benchmarks. Therefore, we chose the best available benchmark for fitness prediction: ProteinGym, specifically its zero-shot setup. It also allows us to evaluate TTT in a different setting compared to the structure and function prediction tasks, where downstream task heads $h$ are fully supervised. We have updated the Evaluation Setup paragraph of Section 4.1 to incorporate this clarifications.
>
> **We have clarified why we did not employ the supervised ProteinGym benchmark.** To the best of our knowledge, unlike the zero-shot ProteinGym benchmark, the supervised ProteinGym benchmark has not yet been widely adopted and currently features only methods from a single publication, ProteinNPT ([Notin et al., 2023](https://www.biorxiv.org/content/10.1101/2023.12.06.570473v1)). According to the official ProteinGym GitHub page, the supervised benchmark is not fully integrated into the ProteinGym codebase and is only accessible through the ProteinNPT codebase. We have now explained it in the ProteinGym paragraph of Section A.1.1 in the Appendix. When experimenting with the ProteinNPT codebase, we encountered computational bottlenecks while reproducing the results of the baseline methods. Specifically, each model needs to be trained on every assay, requiring approximately 24 hours per assay. Unfortunately, after carefully considering the suggestion to evaluate TTT on this benchmark, we found this task infeasible to complete within the rebuttal period.
>
> Conceptually, however, we do not foresee any obstacles preventing TTT from improving the state-of-the-art ProteinNPT model. This model follows the Y-shaped architecture and leverages masked modeling as extensively as the other methods considered in our work. It does so in two ways: first, by utilizing MSA Transformer embeddings and zero-shot predictions as inputs, and second, through the non-parametric transformer approach (NPT), which masks multiple sequences in a batch. To strengthen our argument, we provide results from evaluating MSA Transformer \+ TTT in a zero-shot setting (please see our response to **W2**).
>
> Since ProteinNPT uses MSA Transformer zero-shot predictions and embeddings as inputs, these improvements suggest that TTT could also benefit ProteinNPT when applied. Additionally, TTT can be directly applied to the NPT batch-wide masking step itself, which is expected to further enhance its performance.

---

> > ### Author Response · Authors · 2024-11-22
> >
> > > **W2.** In "Background and Related Work", the authors cite a number of papers that performed model adaptation on either protein MSAs or otherwise related proteins to the target protein. These methods are then dismissed because "constructing MSAs is time-consuming, and similar proteins may not be available for many targets." These arguments are situational and are not strong enough reasons to avoid direct comparisons with the cited methods. For example, one may be making predictions for many closely related target proteins, in which case constructing a single MSA and using that to adapt the model for all target proteins (as in TranceptEVE) may be faster than applying TTT to each target. The paper would be strengthened by including a more in-depth discussion about the distinction between existing methods and the authors' proposed method, a discussion about the situations in which each type of method is relevant, and a quantitative experiment demonstrating improved performance in a task between the proposed method and the existing methods.
> >
> > > * A reasonable comparison is to compare the difference in performance between TranceptEVE and Tranception to the difference in performance between methods w/o TTT to those using TTT (e..g SaProt (650M) and SaProt (650M)+TTT). On avg spearman, the difference between TranceptEVE and Tranception is about 0.02, while the difference between methods with and without TTT is usually about an order magnitude smaller (0.0014 for SAProt (650M) and SAProt (650M)+TTT). This is not a perfect comparison, but it does suggest that the MSA-based adaption of TranceptEVE can have advantages over TTT, which warrants further investigation about the distinction and relative advantages of the methods.
> >
> >   Please kindly note that TTT and evolutionary features are not competing approaches. In this work we have developed a test-time training approach for fine-tuning a model on a single sequence using masked modeling. A similar approach can also be applied to fine-tune a model on the whole MSA using masked modeling (please see response to W4 for the first results in this direction). In the current paper we focused on the single sequence setup because it is conceptually easier and, therefore, more suitable for our work in test-time training for proteins. In many scenarios, using a single sequence is also beneficial compared to using the whole MSA. For example, single-sequence-based test-time training is beneficial in the following cases:
> >
> > - **The computational efficiency of the predictive method is important**. For example, ESMFold+TTT presented in our paper can be applied on large scale to metagenomic data (e.g., to extend ESM Atlas; [Lin et al., 2023](https://www.science.org/doi/10.1126/science.ade2574)), while relying on MSA would make a method an order of magnitude slower and not applicable on the large scale (please see newly added Figure 9 in the Appendix).
> > - **MSA is not available or very poor**. Table 5 in the Appendix demonstrates that TTT is most beneficial for proteins that have poor MSA. This highlights that TTT developed in this work is especially useful where MSA-based methods fail.
> >
> >   Indeed, as reviewer accurately points out using TTT to fine-tune the model on the entire MSA rather than a single sequence can lead to better performance, though it introduces significant computational overhead for building the MSA and may not be applicable for proteins lacking close homologs (e.g., proteins in viruses), as discussed in the previous paragraph. We conducted additional experiments to demonstrate that TTT can be combined with evolutionary information via MSA to further enhance performance. Please note that while experiments with TranceptEVE could indeed be a highly reasonable choice, we perform the experiments in a more simplified and controllable setup to highlight different opportunities of combining TTT with MSA (please see our response to **Q3** for details about our choice).
> >
> > *Please see the continuation below.*

---

> > > ### Author Response · Authors · 2024-11-22
> > >
> > > Indeed, as the reviewer accurately points out using TTT to fine-tune the model on the entire MSA rather than a single sequence can lead to better performance, though it introduces significant computational overhead for building the MSA and may not be applicable for proteins lacking close homologs (e.g., proteins in viruses), as discussed in the previous paragraph. We conducted additional experiments to demonstrate that TTT can be combined with evolutionary information via MSA to further enhance performance. Please note that while experiments with TranceptEVE could indeed be a highly reasonable choice, we perform the experiments in a more simplified and controllable setup to highlight different opportunities of combining TTT with MSA (please see our response to **Q3** for details about our choice).
> > >
> > > **TTT can be used to inject MSA into MSA-agnostic models at test time.** The table below (not yet incorporated into the manuscript) extends Table 5 by including new results for ESM2 (35M) \+ TTT\_MSA and ESM2 (35M) \+ TTT \+ TTT\_MSA. The first method adapts ESM2 (35M) \+ TTT to predict distributions of MSA columns instead of the standard one-hot distributions corresponding to wild types during self-supervised test-time fine-tuning. This allows ESM2 (35M) to adapt to a test protein by incorporating (co)evolutionary features, enabling test-time injection of MSA even though the model was not pre-trained on such features. Results show that this version of test-time training outperforms the original single-sequence TTT on proteins with MSAs of low (2nd column) and medium (3rd column) depth but performs worse on MSAs with high depth (4th column; most likely because ESM2 training dataset contains many homologous sequences and additional fine-tuning using MSA is highly prone to immediate overfitting). This suggests using single-sequence TTT for proteins with rich MSAs and TTT\_MSA for proteins with sparse MSAs. The combined approach, ESM2 (35M) \+ TTT \+ TTT\_MSA, achieves the best performance.
> > >
> > >
> > >   The TTT\_MSA approach can also be applied to any other method discussed in our paper. Additionally, more advanced loss functions for fine-tuning models on MSA, such as EvoRank ([Gong et al., 2024](https://openreview.net/forum?id=XblaAN1jq6)), hold significant promise for even more effective integration of evolutionary information through TTT.
> > >
> > >
> > > | Model | Avg. Spearman ↑ | Spearman (Low MSA depth) ↑ | Spearman (Medium MSA depth) ↑ | Spearman (High MSA depth) ↑ |
> > > | :---- | :---- | :---- | :---- | :---- |
> > > | ESM2 (35M) | 0.3211 | 0.2394 | 0.2707 | 0.4510 |
> > > | ESM2 (35M) \+ TTT\_MSA | 0.3366 | **0.3377** | **0.3430** | 0.3809 |
> > > | ESM2 (35M) \+ TTT | 0.3406 | 0.2445 | 0.3142 | **0.4598** |
> > > | ESM2 (35M) \+ TTT \+ TTT\_MSA | **0.3601** | **0.3377** | **0.3430** | **0.4598** |
> > >
> > >   *Please note that for this experiment we only used one random seed in the limited time frame of the rebuttal, while TTT is stable across different seeds, as shown in other experiments (e.g. Table 1).*
> > >
> > >
> > >   **TTT can be used to boost performance of models trained on MSAs.** Additionally, we implement MSATransformer+TTT, applying TTT to MSATransformer ([Rao et al., 2021](https://www.biorxiv.org/content/10.1101/2021.02.12.430858v3)). Specifically, we apply the same TTT objective function used for a single test sequence, while also incorporating the MSA as input to the model, as done in the MSA Transformer. The table below (not yet incorporated into the manuscript) shows that, even without tuning hyperparameters (reused from experiments with ESM2), MSATransformer+TTT outperforms MSATransformer.
> > >
> > >
> > > | Model | Avg. Spearman ↑ |
> > > | :---- | :---- |
> > > | MSA Transformer | 0.4206 |
> > > | MSA Transformer \+ TTT | **0.4223** |
> > >
> > >   *Please note that for this experiment we only used one random seed in the limited time frame of the rebuttal, while TTT is stable across different seeds, as shown in other experiments (e.g. Table 1). Please also note that for the same reason we do not use an ensemble with 5 random seeds as done in ProteinGym.*

---

> > > > ### Author Response · Authors · 2024-11-22
> > > >
> > > > > **W3.** The result on improving structure prediction in ESM3 and ESMFold is interesting, but of limited practical relevance since AlphaFold2/3 are far superior structure predictors. The paper would be strengthened by either demonstrating an application of TTT to AlpahFold or an explanation of why this is not possible.
> > > >
> > > > **We have clarified the high practical relevance of applications of TTT to ESMFold and ESM3 supported by new experimental analysis.** Indeed, AlphaFold2/3 are far superior structure predictors compared to ESM3 and ESMFold in terms of accuracy. Nevertheless, ESMFold and ESM3 are far superior in terms of computational and memory demands and are widely adopted by the community for their ease of use. For example, ESMFold enabled structural characterization of large metagenomics data (\>617 million metagenomic sequences), which would be infeasible with AlphaFold. We have conducted an additional analysis (new Figure 9 in the updated manuscript) providing evidence that ESMFold+TTT remains in the same category of fast (as opposed to highly accurate) methods, being significantly faster than AlphaFold. At the same time, ESMFold+TTT enables more accurate predictions compared to the original ESMFold. For example, the original ESMFold has high confidence predictions only for 36% of sequences from the metagenomic database, while the other 392 million sequences remain with low or medium confidence predictions. Applying ESMFold+TTT to these remaining sequences could significantly expand the metagenomic atlas characterized by ESMFold (Fig 10 in the Appendix of the updated paper suggests that ESMFold+TTT significantly improves predictions in almost 40% of challenging sequences).
> > > >
> > > > **We have clarified that application of TTT to AlphaFold is also possible.** Since AlphaFold2 was trained using masked language modeling among the loss terms, one way to leverage TTT is to first fine-tune the method’s backbone on a test sequence (or whole MSA) using masked modeling and then predict the structure. Unfortunately, implementing and evaluating AlphaFold2+TTT would be highly resource-intensive, and especially time-consuming in the limited timeframe of this rebuttal. We hope that our previous paragraph clarifies why the application of TTT to ESMFold and ESM3 is equally, if not more, practically relevant than its application to AlphaFold, particularly in scenarios where computational efficiency and scalability are important.
> > > >
> > > > > **W4.** Less important: the authors claim in section 3.3 to offer an explanation for the effectiveness of test time adaptation by appealing to the correlation between perplexity minimization and model performance. This is somewhat circular, since the proposed method directly lowers the perplexity of the target protein. Therefore, the positive results provide mre evidence that perplexity minimization is important, but don't provide an explanation for why the method works. I recommend framing this as a justification for the method rather than an explanation for why it works.
> > > >
> > > > **We appreciate the reviewer’s comment and have reframed the section as a justification for TTT effectiveness.** Please note that the method **does not directly minimize perplexity**, instead it minimizes the cross-entropy loss. The reasoning of Section 3.3 can be summarized in the following chain of implications: lower cross-entropy \-\> lower perplexity \-\> higher downstream task performance, justifying the increased downstream performance upon cross-entropy minimization. In detail, TTT minimizes the cross-entropy loss of true tokens (i.e., wild-type amino acids), which reduces perplexity (as theoretically shown in the cited reference [(Hennigen & Kim, 2023)](https://arxiv.org/abs/2305.15501)), and reduced perplexity, in turn, leads to better downstream performance (as shown empirically through correlation in our paper). Thus, minimizing cross-entropy in the self-supervised task indirectly enhances performance in the downstream supervised task. This linkage is mediated through perplexity minimization, as discussed in detail in Section 3.3. We hope this clarification resolves any misunderstandings.

---

> > > > > ### Author Response · Authors · 2024-11-22
> > > > >
> > > > > > **Q1.**  Why did the authors choose to evaluate on the zero-shot tasks in ProteinGym, rather than the fine-tuning tasks? Is there literature precedent for using test time adaption to improve zero-shot prediction?
> > > > >
> > > > > The aim of our paper is to demonstrate the potential of TTT across well-established problems and benchmarks. Therefore, **we chose the zero-shot task from ProteinGym because it is the best established benchmark for fitness prediction**. In contrast, the fine-tuning task from ProteinGym is not yet widely adopted by the community, with no public code (still in the process of migration from another repository into ProteinGym) and no baselines implemented besides those implemented by the ProteinGym authors. Please see the details  in W1 above. We are not aware of literature discussing the benefits of zero-shot setup in the context of test-time adaptation.
> > > > >
> > > > > > **Q2.** What is the rough time scaling of TTT? How does it compare to the scaling of MSA generation?
> > > > >
> > > > > **We have included an additional analysis (Figure 9 in Appendix) to highlight the high scalability of TTT, showing that TTT is an order of magnitude faster than MSA generation**. Furthermore, TTT directly produces predictions, whereas MSA serves only as a feature, requiring additional processing through a neural network (e.g., AlphaFold2). Notably, even without considering the time required for MSA generation, the forward pass through AlphaFold2 alone takes more time than the entire ESMFold \+ TTT pipeline with 30 steps. This underscores the efficiency of TTT as a practical alternative in scenarios where computational speed and scalability are important.
> > > > >
> > > > > > **Q3.** Can the authors demonstrate a scenario in which TTT clearly outperforms the MSA-based adaptation of TranceptEVE?
> > > > >
> > > > > We add two new experiments in this direction discussed in **W2**. Please note that our goal with TTT is not to compete with MSA. Instead, two approaches can be combined. To illustrate this, we present two simplified and more accessible setups rather than combining TTT with TranceptEVE, though such a combination is indeed feasible since Tranception uses masking-based pre-training. Experimenting with TranceptEVE during the rebuttal phase would be computationally intensive and would require extending our method from random (bidirectional) masking to autoregressive masking, necessitating additional research. Furthermore, since Tranception already incorporates evolutionary information via “inference-time retrieval,” comparing Tranception+TTT with TranceptEVE may not effectively highlight the differences between TTT and MSA.
> > > > >
> > > > > > **Q4.** Is it possible to apply TTT to AlphaFold or OpenFold?
> > > > >
> > > > > Yes, it is possible to apply TTT to both AlphaFold and its reimplementation OpenFold. Please see our comment on AlphaFold in **W3** above.

---

> > > > > > ### Comment · Reviewer_Rmaa · 2024-11-26
> > > > > >
> > > > > > I thank the reviewers for their effort in responding to my comments. However, my primary concern about the comparison of TTT to existing MSA-based adaptation methods has not been sufficiently addressed. In particular, the authors have avoided direct comparison with TranceptEVE and instead compared to a version of TTT adapted to use MSAs; this version of TTT may be suboptimal compared to the more established method of TranceptEVE and thus does not represent a strong baseline to compare to MSA-based methods. Without comparing Tranception, TranceptEVE, Tranception+TTT and TranceptEVE+TTT, I do not think there is strong support for making the conclusion that TTT can be used in conjunction with other adaptation methods and yield significantly improved results compared to the results using MSA adaptation alone.

---

> > > > > > > ### Author Response · Authors · 2024-12-02
> > > > > > >
> > > > > > > Thank you for your comment, your engagement in the discussion, and for pointing us to these works.
> > > > > > >
> > > > > > > We thoroughly reviewed the methods suggested (Tranception, TranceptEVE, Tranception+TTT, and TranceptEVE+TTT). However, reproducing and implementing all four methods proved challenging within the limited rebuttal period due to their complexity. We invested significant effort into understanding their codebases and design, but the scope of implementation exceeded the time available. Unfortunately, the training code for both Tranception and TranceptEVE does not seem to be available (at least we could nof find it), which complicates their combination with TTT, considering that the models follow autoregressive masking, as opposed to bi-directional masking employed for other models we studied in the paper. This also requires extending the theoretical analysis (Section 3.3) to autoregressive models.
> > > > > > >
> > > > > > > Instead, we focused on demonstrating the effectiveness of TTT for fitness prediction using ESM2, MSA Transformer and SaProt. Importantly, SaProt achieves state-of-the-art performance on the ProteinGym benchmark and is not straighforward to combine with MSA, making it an ideal candidate to showcase the unique benefits of TTT and its extension to MSA (TTT_MSA). We showed that TTT improves SaProt, which already surpasses TranceptEVE on ProteinGym. Additionally, our experiments with ESM2 + TTT_MSA and MSA Transformer + TTT illustrate that TTT and MSA can be effectively combined (please see our response to W2). While we agree with the reviewer that applying TTT to Tranception and TranceptEVE could strengthen our argument, we hope our consistent findings—showing that TTT enhances performance across diverse models (ESM2, MSA Transformer, SaProt, ESMFold, ESM3, TerpeneMiner, Light Attention), their scales (35M to 3B parameters), and benchmarks (ProteinGym, MaveDB, CAPRI, TPS, setHard)—strongly suggest that TTT would likely enhance other models as well, including Tranception and TranceptEVE.
> > > > > > >
> > > > > > > Finally, we would like to note that TTT and MSA are not inherently competitive methods; each has its own strengths and weaknesses. TTT is particularly valuable when MSA is unavailable or poor (please see Table 5 in Appendix B.1). Moreover, TTT is especially beneficial in cases where MSA is hard to leverage  for model adaptation. For instance, it is unclear how to incorporate MSA with the state-of-the-art fitness predictor SaProt. Here our solution (TTT_MSA) provides a simple and effective way to achieve this.

---

### Official Review · Reviewer_QVrn · 2024-11-03

**Soundness:** 2
**Presentation:** 3
**Contribution:** 2
**Rating:** 3
**Confidence:** 3

**Summary:**

Data scarcity and distribution shifts impede generalization in machine learning models for biological data. To overcome this, this paper proposes a self-supervised fine-tuning method at test time, which adapts models to specific proteins without extra data, enhancing generalization and achieving state-of-the-art results in protein fitness prediction.

**Strengths:**

Good attempt at biological problems.

**Weaknesses:**

**Section 2 - Model Adaptation:**
- **TTT Method Selection:** The rationale for selecting the current TTT methods is unclear. Provide a clear justification for the choice.

**Section 3 - Training Objectives:**
- **Performance Linkage:** The objectives should be linked to improved biological performance. Add relevant goals to achieve desired outcomes.
- **Efficiency:** Reducing perplexity is insufficient for justifying large model use. Considering smaller discriminators or LLM query filtering for efficiency is the same.
- **Resource Management:** Finetuning large models is resource-intensive. Assess cost-effectiveness and seek more efficient training strategies.

**Questions:**

see weaknesses

---

> ### Author Response · Authors · 2024-11-22
>
> > **W1.** TTT Method Selection: The rationale for selecting the current TTT methods is unclear. Provide a clear justification for the choice.
>
> We have modified the opening paragraph of Section 2 to clarify that the section provides the rationale, feasibility, and broad applicability of the TTT method. We have also improved the overall clarity of our text to better highlight the justification of our method.
>
> > **W2.** Performance Linkage: The objectives should be linked to improved biological performance. Add relevant goals to achieve desired outcomes.
>
> We have extended the motivation for applying TTT to protein structure prediction by discussing its biological implications in Section 4.2:
>
> - “Arguably, one of the most remarkable applications of machine learning in the life sciences has been in protein folding (Jumper et al., 2021; Lin et al., 2023; Abramson et al., 2024), paving the way for numerous advances in the understanding of biology (Yang et al., 2023; Akdel et al., 2022; Barrio-Hernandez et al., 2023).”
>
> The objectives for developing TTT for proteins are currently outlined in the Introduction, while the applications of TTT to protein fitness, structure, and function prediction are linked to improved biological performance in the opening paragraphs of Sections 4.1, 4.2, and 4.3, respectively:
>
> - “Bridging the gap between broad, dataset-wide optimizations and the precision required for studying single proteins in practical applications remains a critical challenge in integrating machine learning into biological research (Sapoval et al., 2022). … Our method enables adapting protein models to one protein at a time, on the fly, and without the need for additional data.”
> - “Protein fitness refers to the ability of a protein to efficiently perform its biological function, which is determined by its structure, stability, and interactions with other molecules. … In this paper, we demonstrate that applying test-time training (TTT) to representative models, such as ESM2 (Lin et al., 2023\) and SaProt (Su et al., 2023), enhances their protein fitness prediction capabilities.”
> - “Protein function prediction is essential for understanding biological processes and guiding bioengineering efforts. … Therefore, enhancing generalization through TTT is particularly relevant in this context.”
>
> We are open to extending the discussion of our goals to include any additional specific objectives that might be suggested.
>
> > **W3.** Efficiency: Reducing perplexity is insufficient for justifying large model use. Considering smaller discriminators or LLM query filtering for efficiency is the same.
>
> It is a valid point that reducing perplexity is insufficient for justifying large model use. Please note that the goal of our method is to adapt a given protein LLM, regardless of its size, to a specific protein sequence, with perplexity reduction as the means to achieve this. We focus on large protein language models to demonstrate the applicability of our method to state-of-the-art models, which are typically large, rather than limiting results to smaller, proof-of-concept models. We have clarified this aspect in the paragraph “Fine-tuning large models” in Section 3.1:
>
> - “Since state-of-the-art models for many protein-oriented tasks are typically large, with up to billions of parameters, our aim presents two key challenges… To address the first challenge, we perform forward and backward passes through a small number of training examples and accumulate gradients… We address the second challenge by employing low-rank adaptation…”
>
> > **W4.** Resource Management: Finetuning large models is resource-intensive. Assess cost-effectiveness and seek more efficient training strategies.
>
> Thank you for pointing this out. We have added a new Figure 9 in the Appendix highlighting the cost-effectiveness of our TTT approach using low-rank adaptation (LoRA) and gradient accumulation. We have referenced it in the main text:
>
> - “Figure 9 in Appendix B shows that ESMFold \+ TTT maintains computational efficiency comparable to ESMFold while being orders of magnitude faster than AlphaFold2.”

---

> > ### Comment · Reviewer_QVrn · 2024-11-25
> > **Providing a comparative analysis of the strengths and limitations of different TTT methods.**
> >
> > I commend your efforts to apply Test-Time Training (TTT) to protein language models. However, I have a few specific comments and suggestions that I hope you will consider in your revision.
> >
> > First, “the rationale for choosing the current TTT approach is unclear.“ means elaborating on the reasoning behind selecting the specific TTT approach and providing a comparative analysis of the strengths and limitations of different TTT methods. As shown in "Revisiting Realistic Test-Time Training: Sequential Inference and Adaptation by Anchored Clustering," TTT can be categorized into one-pass adaptation and multi-pass adaptation. Your paper does not sufficiently compare various methods within the field, instead relying on a straightforward application as "Test-Time Training with Masked Autoencoders." To enhance the depth and breadth of your work, I recommend elaborating on the reasoning behind selecting the specific TTT approach and providing a comparative analysis of the strengths and limitations of different TTT methods.
> >
> > Second, the advantage of TTT in NLP lies in its ability to continuously learn during testing, thereby improving long sequence or contextual handling. However, protein datasets do not inherently possess contextual relationships, raising questions about whether TTT is suitable for protein data analysis, especially given its similarity to fine-tuning. I suggest further discussing the applicability of TTT to protein data, along with providing relevant theoretical or experimental evidence to support your choice.

---

> > > ### Author Response · Authors · 2024-11-25
> > >
> > > We highly appreciate the reviewer’s engagement in the discussion. Below, we address the reviewer’s valuable comments one by one and confirm that all clarifications and citations will be incorporated into the manuscript.
> > >
> > > > 1. On “the rationale for choosing the current TTT approach is unclear.“
> > >
> > > This is an important point. We provide the rationale for using the current TTT approach by contextualizing it within the comprehensive categorization provided in the paper kindly shared by the reviewer, ([Su et al., 2023](https://arxiv.org/abs/2303.10856)). First, please note that our central motivation for developing TTT for proteins is that practitioners (e.g., biologists) are often most interested in accurate predictions for individual proteins they study, rather than for batches or entire test datasets. In practical applications, performance on single proteins is frequently prioritized over performance averaged across a test set or batches of proteins.
> > >
> > > **Given this practical motivation, the TTT approach chosen in our work is specifically suitable.** Specifically, unsupervised domain adaptation (UDA) methods ([Ganin & Lempitsky, 2015](https://arxiv.org/pdf/1409.7495); [Tzeng et al., 2014](https://arxiv.org/pdf/1412.3474)) require access to the entire test dataset during training, which is not feasible in our scenario. For example, a practitioner might need to predict the fitness of a completely novel protein in the context of *in silico* directed evolution ([Kirjner et al., 2023](https://arxiv.org/abs/2307.00494)) or predict the structure of *de novo* generated proteins absent from any database ([Watson et al., 2023](https://www.nature.com/articles/s41586-023-06415-8)).
> > >
> > > Similarly, source-free domain adaptation (SFDA) methods are also unsuitable, as they typically assume access to either the entire test dataset or batches of test data points ([Liu et al., 2021](https://openreview.net/forum?id=86NHK__yFDl); [Liang et al., 2021](https://arxiv.org/pdf/2002.08546)). In contrast, the test-time training (TTT) paradigm in computer vision assumes access to a single test input at a time, aligning well with our practical motivation. Specifically, our TTT adapts models to one protein at a time (i.e., **one-pass adaptation**) without modifying the pre-training loss function. This places our method in the category of sequential test-time training (sTTT), as classified in [Su et al., 2023](https://arxiv.org/abs/2303.10856), where it is described as "the most realistic TTT protocol."
> > >
> > > Indeed, our work is inspired by “Test-Time Training with Masked Autoencoders” ([Gandelsman et al., 2022](https://arxiv.org/abs/2209.07522)), which applies TTT to vision transformers to address explicit distribution shifts in image classification. However, our work is not a direct application of the former and required significant design choices to adapt TTT to proteins (please refer to Section 3 for details). Please also note that we do not exclude the possibility of other test-time adaptation paradigms being suitable for proteins and we highlight several in our Discussion section, where we will also add TTAC++, kindly shared by the reviewer. We selected the current TTT due to its simplicity and suitability for the initial work on test-time adaptation for proteins, as also acknowledged by Reviewer dg8u. We welcome suggestions from the reviewer regarding alternative TTT frameworks we may not have considered.
> > >
> > > > 2. On the relevance of TTT for proteins
> > >
> > > We demonstrate that our TTT method consistently improves performance across different protein models, scales, and datasets. Specifically, we address three well-established problems and use state-of-the-art benchmarks (Section 4). These experimental results strongly support the suitability of TTT for proteins.
> > >
> > > From a theoretical perspective, we link our TTT method to perplexity minimization in protein language models. We show that our method effectively minimizes perplexity on test proteins, a metric empirically known to correlate with better downstream performance (Section 3.3).
> > >
> > > We assume that by “the advantage of TTT in NLP lies in its ability to continuously learn during testing, thereby improving long sequence or contextual handling,” the reviewer refers to [Sun et al., 2024](https://arxiv.org/html/2407.04620v1), where TTT is applied to model long sequences with contextual handling. However, TTT applications in NLP are not limited to this scenario and have been used to adapt to test queries regardless of their length ([Hardt & Sun, 2024](https://openreview.net/forum?id=CNL2bku4ra );[Hübotter, 2024](https://arxiv.org/pdf/2410.08020)). Additionally, modeling long sequences with transformers is of significant practical interest in protein machine learning ([Sgarbossa et al., 2024](https://www.biorxiv.org/content/10.1101/2024.05.24.595730v1)).
> > >
> > > Please let us know if there are any other points requiring clarification, or kindly consider raising the score if there are no further concerns.

---

### Official Review · Reviewer_dg8u · 2024-11-03

**Soundness:** 3
**Presentation:** 3
**Contribution:** 3
**Rating:** 6
**Confidence:** 3

**Summary:**

The authors present a technique for self-supervised fine-tuning during the testing phase, enabling models to dynamically adapt to the specific test protein without extra data. Their experiments show that the approach achieves new state-of-the-art results on three tasks: (1) standard benchmarking for protein fitness prediction, (2) enhanceing protein structure prediction, and (3) improving protein function predictions. They also establish a connection between their objective and perplexity minimization.

**Strengths:**

## Motivation

The paper is well-motivated. There is an abundance of popular, large, and expensively-trained models by major institutions on massive amounts of protein data that perform "subpar" at best on specific downstream tasks. This under-performance problem has been established time and time again in subsequent studies and paper. Any simple approach that can mitigate this problem, however modestly, can positively influence the future work in this area. This paper is addressing this problem directly.

The proposed approach avoids being overly complex, and it seems plausible to generalize to downstream tasks not included in the data. The method does not require new labels, an auxillary dataset, or prior knowledge about the tasks. I belive these features make this study well-placed in the literature.

## Supporting the Claims

The main experiments section provides enough data to satisfy the three claimed tasks. The experiments prior to the main section (e.g., Figures 3 and 4) substantiate the claims made in the introduction and model description section. Since the contributions in the paper are simple enough to verify (e.g., the correlation of the perplexity measure and the TTT approach), I find the improvements related to the contribution of the approach.

## Strengths

1. The paper is quite well-written. I enjoyed reading the paper, understanding the approach and the problem, and the explanations were easy on the mind. The key contributions were stated clearly in the introduction, and the abstract and the title are fair representatives of the work.

2. The experimentation quality is reasonable and statistically significant confidence intervals were reported. That being said, the number of random seeds is a bit on the lower side (i.e., 5 is not too large).

3. The appendix contains notable valuable and complementary information to the paper; (1) many ablation studies and results for a number of hyper-parameters were included, and (2) the experimental setup is clearly detailed.

4. The improvements seem modest but consistent across the board, for instance in Table 1. Even when the proposed method does not work best, it doesn't perform terrible. It is relieving to see this approach does not induce a lot of risk.

**Weaknesses:**

My main concern is the proper estimation of the number of test-time training steps. Such fine-tunings have been studied frequently and are prevalant in other areas, such as few-shot learning, meta-learning, domain adaptations, etc. Most, if not always, (1) the size of the fine-tuned parameters, and (2) the number of training steps can play a significant role in the outcome.

While the paper is not devoid of studies regarding these effects, I don't feel they are complete enough. Once would expect a clear degradation in performance with large-enough step sizes and model-complexity. Some intuitive indications of when and how TTT should be proposed, studied, and tested. There is almost no theoretical component to the paper, which is not a deal breaker for this paper, but it may assist in the correct choice of such hyper-parameters.

A more thorrough discussion and experimental results regarding the **limitations** of the approach is also missing from the paper.

## Recommendation

Based on the rigor of the studies, the narrowness of the claims, and the quality of the writing, I'm suggesting acceptance. IMHO, this paper is in a closer shape to being published than being initially drafted, and strikes me in the upper percentiles of quality among past ICLR papers. That being said, I still would like to see my main concern being addressed (see the weaknesses section).

**Questions:**

The improvements in structure prediction seem slightly more significant than the fitness or function prediction tasks. Would you care to comment on why this may be the case?

## Minor Comments

1. Line 54: seems unnecessarily cut off.

2. Line 477: "Evaulation" in "Evaulation setup" should be "Evaluation"​.

3. Line 508: "syntase" in "terpene syntase (TPS)" should likely be "synthase"​.

4. Inconsistent Hyphenation: Terms like "protein language model" and "test-time training" sometimes appear with inconsistent hyphenation (e.g., "test time-training")​.

---

> ### Author Response · Authors · 2024-11-22
>
> > **W1.** My main concern is the proper estimation of the number of test-time training steps. Such fine-tunings have been studied frequently and are prevalant in other areas, such as few-shot learning, meta-learning, domain adaptations, etc. Most, if not always, (1) the size of the fine-tuned parameters, and (2) the number of training steps can play a significant role in the outcome.
>
> > While the paper is not devoid of studies regarding these effects, I don't feel they are complete enough. Once would expect a clear degradation in performance with large-enough step sizes and model-complexity. Some intuitive indications of when and how TTT should be proposed, studied, and tested. There is almost no theoretical component to the paper, which is not a deal breaker for this paper, but it may assist in the correct choice of such hyper-parameters.
>
> We thank the reviewer for the thoughtful comments, which we address below.
>
> **The number of training steps**
>
> For each task, we used held-out validation data to fine-tune three primary hyperparameters: learning rate, batch size (or the number of gradient accumulation steps), and the number of TTT steps. The same final task-specific set of hyperparameters was applied to each target test protein, and the overall results were reported. As shown in Table 4 in the Appendix, we experimented with different learning rates and batch sizes (“Grad. acc. steps”) while keeping the number of TTT steps fixed. The rationale for fixing the number of steps is illustrated in Figure 10, which demonstrates that—despite having three hyperparameters—there are effectively only two degrees of freedom controlling the performance of the model. In other words, by keeping the number of steps constant, the expected performance can be controlled by adjusting the learning rate and the batch size. We have clarified this observation in the caption of Figure 10\. We have also clarified the caption of Table 4 to highlight that, for this reason, we used 30 TTT steps in all our experiments. The only exception was ESM3 \+ TTT, where the number of steps was inconsistently set to 50 during initial experiments with different models and tasks conducted in parallel before standardizing the number of steps to 30\.
>
> For the task of protein structure prediction, we determined the optimal number of steps for each sequence using a well-established measure of model confidence: the pLDDT confidence score of each residue predicted in the structure, averaged across the sequence. We have added a new Figure 11 in the Appendix, demonstrating that, with the final learning rate and batch size, the number of steps could be reduced to improve computational efficiency while still achieving a significant improvement in performance. However, as shown in the newly added Figure 9 in the Appendix, our approach remains efficient even with 30 steps.
>
> In future work, we aim to explore other heuristics for estimating the number of steps, focusing on methods that assess the quality of protein representations from a protein language model and can be applicable to any downstream task, such as linear probing of protein contact maps ([Rives et al., 2020](https://www.pnas.org/doi/10.1073/pnas.2016239118)) or the recently proposed categorical Jacobian ([Zhang et al., 2024](https://www.pnas.org/doi/10.1073/pnas.2406285121)).
>
> **The size of the fine-tuned parameters**
>
> Our work suggests the following relationships between the number of fine-tuned parameters and the performance of TTT:
>
> - **TTT benefits smaller models more than larger models.**  This is supported by Table 5 and Table 6 in the Appendix, which compare the performance gains from applying TTT to ESM2 (35M) and ESM2 (650M), as well as SaProt (35M) and SaProt (650M). Across the two datasets and two models, TTT consistently results in larger improvements on smaller models.
> - **LoRA is only beneficial when the backward pass through the full model does not fit into GPU memory.** We have extended Table 4 in the Appendix to show that we experimented with applying LoRA to ESM2 (35M) and ESM2 (650M), both of which can be fine-tuned without it. In line with the original LoRA paper ([Hu et al., 2021](https://arxiv.org/abs/2106.09685)), we did not observe any improvements in downstream task metrics when using LoRA in these cases.
> - **Fine-tuning all layers outperforms fine-tuning a subset of layers.** When predicting protein structures, the ESMFold head $h$ extracts information from the ESM2 backbone $f$ by computing a learnable linear combination of embeddings across all layers. During initial experiments with ESMFold \+ TTT, we observed that 92% of the weights are attributed to the final layer and 5% to the second layer. Motivated by these findings, we experimented with fine-tuning only these two layers or a subset of the last layers in various combinations. However, the performance never surpassed the performance achieved when fine-tuning all layers.

---

> > ### Author Response · Authors · 2024-11-22
> >
> > > **W2.** A more thorrough discussion and experimental results regarding the limitations of the approach is also missing from the paper.
> >
> > We have expanded the Discussion section to highlight the limitations of our work. Throughout the main text we have also better referenced the figures in the Appendix showing the limitations of our method. In summary, the limitations include (i) a lack of comprehensive understanding of why TTT is able to enhance model generalization, (ii) the identification of its success and failure modes, and (iii) the need for careful hyperparameter tuning when model confidence estimates are unavailable. Figure 10 in the Appendix quantifies the number of success and failure cases when applying TTT to protein structure prediction, while Figure 13 demonstrates that TTT may fail with inappropriately chosen hyperparameters.
> >
> > > **Q1.** The improvements in structure prediction seem slightly more significant than the fitness or function prediction tasks. Would you care to comment on why this may be the case?
> >
> > The more significant improvement in structure prediction can be attributed to the availability of a reliable estimate of prediction quality, which allows effective control over TTT progress. Among the tasks studied in this paper, structure prediction is the most mature, with state-of-the-art predictors providing pLDDT (predicted Local Distance Difference Test) as a reliable measure of prediction quality. At each step of TTT, we predict pLDDT and select the step with the highest value for the final prediction. This strongly mitigates the risk of overfitting. The newly added Figure 11 in the Appendix illustrates that any number of TTT steps leads to improvement, while Figure 14 demonstrates the robustness of ESM3 \+ TTT to the choice of hyperparameters. Therefore, for structure prediction, TTT is only likely to have a negative impact if pLDDT is inaccurate. However, pLDDT prediction in ESMFold and ESM3 used in this work are well-calibrated. In contrast, the absence of a confidence estimate can introduce a risk of overfitting during TTT. For instance, as shown in new Figure 8, the fitness prediction model begins to overfit after step 10\. A reliable predicted quality estimate would prevent this by selecting step 10 as the final prediction, thereby avoiding overfitting.
> >
> > > **Minor Comments**
> >
> > We thank the reviewer for highlighting the typos and formatting inconsistencies. We have fixed all of them in the text.

---

> ### Comment · Reviewer_dg8u · 2024-12-02
> **Response to Rebuttals**
>
> I would like to thank the authors for addressing my comments and keep my score as it was.

---

### Official Review · Reviewer_UQtJ · 2024-11-06

**Soundness:** 3
**Presentation:** 3
**Contribution:** 3
**Rating:** 5
**Confidence:** 3

**Summary:**

In this study, training/fine tuning at test time (TTT) is applied to adopt the representations learned by foundation models, trained on large corpus of protein families, to a specific task defined on a test protein. The proposed technique has been extensively benchmarked to demonstrate improvements in protein fitness, structure and function prediction tasks.

**Strengths:**

1. Proposes a general technique that can be used with any backbone trained with masked language model head. However, the idea is not novel.

2. The paper has a nice “Related Work” section as well as clear and easy-to-follow narrative.

**Weaknesses:**

1. As acknowledged by the authors, the TTT technique has been used in other domains, however this study applies it for the first time to the computational problems in protein domain.

2. One of the motivations of this work is to get rid of additional data collection (MSA) for a given test protein, however the experiments do not show this in a comparable setting. I may be wrong, so the author’s clarification is appreciated. Please see the questions for more detailed on this.

    **(i)** It is shown that the performance is improved by applying TTT to foundation models, e.g. ESM2. However, there is no comparable setting in which the foundation model is fine-tuned on the sequences homologous to the test protein. Therefore, it is not clear when one should choose to go with TTT instead of MSA, i.e., incorporating evolutionary features.

    **(ii)** It looks like incorporating (co)evolutionary features through fine tuning may be a better option than applying TTT to $fog$ (transformer encoder and masked language model head) for protein fitness prediction. Three out of five phenotypes in Table 1 support this claim.

**Questions:**

1. In Table 1, is the TTT done per protein individually or simultaneously for all proteins in the test set? Does the same applies to the other benchmarked methods as well?

2. Since TTT on one test protein is motivated by the derivation of MSAs to be time consuming, have you tested variants of ESM2 and SaProt where the models ($fog$) are finetuned by the homologous sequences derived for each protein? Do these variants outperform TTT fine tuned models? My understanding is that ESM2 and SaProt see similar proteins to the test/target protein in their training but are not further fine tuned on the MSA derived per protein.

   (i) Also, it would have been nice to see a combination of MSA and TTT, where TTT is done for all the homologous sequences to the test      protein. It is not clear to me whether it hurts or improves the performance compared to only using TTT on the single test protein.

    (ii) The point of homologous sequences can be applied to the other included benchmarks too.

3. What is meant by TTS steps?

4. Is there any preference on what $h$ (task specific head) has been trained on? Should it be the same dataset as the one used for backbone ($f$) training, e.g., like the structure prediction task using ESMFold (Section A.2.3), or it could be any other large dataset with information on the task.

---

> ### Author Response · Authors · 2024-11-22
>
> > **W1.** As acknowledged by the authors, the TTT technique has been used in other domains, however this study applies it for the first time to the computational problems in protein domain.
>
> Thank you for this observation. We acknowledge that the TTT technique has been used in other domains, and we cite the relevant papers in the Background and Related Work sections. Building on ideas from other areas of machine learning has led to powerful methods in the past. Examples include protein language models derived from natural language models or protein structure diffusion models inspired by generative models in computer vision. However, we would like to point out that in order to extend TTT to the protein domain we had to address several challenges, as described below.
>
> - In computer vision, TTT primarily addresses explicitly known distribution shifts (e.g., blurred or rotated images) ([Sun et al., 2020](https://arxiv.org/abs/1909.13231), [Gandelsman et al., 2022](https://arxiv.org/abs/2209.07522)), whereas our work focuses on unknown distribution shifts, which are common when applying machine learning to understudied proteins.
> - While the reasons for TTT’s effectiveness are often unclear in other domains (e.g., in computer vision), we provide a possible explanation in the domain of proteins through perplexity minimization (Section 3.3).
> - Making TTT applicable to a single new unseen target protein on-the-fly required important design choices not addressed in the original TTT papers in computer vision. This includes the use of LoRA (Section 3.1), gradient accumulation (Section 3.1), and the construction of a new dataset from MaveDB to rigorously optimize hyperparameters (Section A.1.1).
>
> To the best of our knowledge, our work represents the first application of the TTT approach to biological data, where adaptation to individual examples is particularly important, as discussed in the first paragraph of the Introduction and also recognized by Reviewer dg8u.
>
>  > **W2.** One of the motivations of this work is to get rid of additional data collection (MSA) for a given test protein, however the experiments do not show this in a comparable setting. I may be wrong, so the author’s clarification is appreciated. Please see the questions for more detailed on this.
>
>
>   We would like to kindly clarify that the motivation of our work is not to remove the need for additional data collection but rather to provide a universal customization method that does not require additional data. This makes our method orthogonal to the availability of additional data, such as MSAs, allowing it to be combined with such data, if available. We demonstrate the possibility of combining our approach with MSA in new experiments described in the following comments.
>
>
>  > **W3.** (i) It is shown that the performance is improved by applying TTT to foundation models, e.g. ESM2. However, there is no comparable setting in which the foundation model is fine-tuned on the sequences homologous to the test protein. Therefore, it is not clear when one should choose to go with TTT instead of MSA, i.e., incorporating evolutionary features.
>
>
>   Please kindly note that TTT and evolutionary features are not competing approaches. In this work we have developed a test-time training approach for fine-tuning a model on a single sequence using masked modeling. A similar approach can also be applied to fine-tune a model on the whole MSA using masked modeling (please see response to W4 for the first results in this direction). In the current paper we focused on the single sequence setup because it is conceptually easier and, therefore, more suitable for our work in test-time training for proteins. In many scenarios, using a single sequence is also beneficial compared to using the whole MSA. For example, single-sequence-based test-time training is beneficial in the following cases:
>
> - **The computational efficiency of the predictive method is important**. For example, ESMFold+TTT presented in our paper can be applied on large scale to metagenomic data (e.g., to extend ESM Atlas; [Lin et al., 2023](https://www.science.org/doi/10.1126/science.ade2574)), while relying on MSA would make a method an order of magnitude slower and not applicable on the large scale (please see newly added Figure 9 in the Appendix).
> - **MSA is not available or very poor**. Table 5 in the Appendix demonstrates that TTT is most beneficial for proteins that have poor MSA. This highlights that TTT developed in this work is especially useful where MSA-based methods fail.

---

> > ### Author Response · Authors · 2024-11-22
> >
> > > **W4.** (ii) It looks like incorporating (co)evolutionary features through fine tuning may be a better option than applying TTT to fog (transformer encoder and masked language model head) for protein fitness prediction. Three out of five phenotypes in Table 1 support this claim.
> >
> >  Indeed, using TTT to fine-tune the model on the entire MSA rather than a single sequence can lead to better performance, though it introduces significant computational overhead for building the MSA and may not be applicable for proteins lacking close homologs (e.g., proteins in viruses), as discussed in the previous paragraph. We conducted additional experiments to demonstrate that TTT can be combined with evolutionary information via MSA to further enhance performance.
> >
> >  **TTT can be used to inject MSA into MSA-agnostic models at test time.** The table below (not yet incorporated into the manuscript) extends Table 5 by including new results for ESM2 (35M) \+ TTT\_MSA and ESM2 (35M) \+ TTT \+ TTT\_MSA. The first method adapts ESM2 (35M) \+ TTT to predict distributions of MSA columns instead of the standard one-hot distributions corresponding to wild types during self-supervised test-time fine-tuning. This allows ESM2 (35M) to adapt to a test protein by incorporating (co)evolutionary features, enabling test-time injection of MSA even though the model was not pre-trained on such features. Results show that this version of test-time training outperforms the original single-sequence TTT on proteins with MSAs of low (2nd column) and medium (3rd column) depth but performs worse on MSAs with high depth (4th column; most likely because ESM2 training dataset contains many homologous sequences and additional fine-tuning using MSA is highly prone to immediate overfitting). This suggests using single-sequence TTT for proteins with rich MSAs and TTT\_MSA for proteins with sparse MSAs. The combined approach, ESM2 (35M) \+ TTT \+ TTT\_MSA, achieves the best performance.
> >
> >  The TTT\_MSA approach can also be applied to any other method discussed in our paper. Additionally, more advanced loss functions for fine-tuning models on MSA, such as EvoRank ([Gong et al., 2024](https://openreview.net/forum?id=XblaAN1jq6)), hold significant promise for even more effective integration of evolutionary information through TTT.
> >
> > | Model | Avg. Spearman ↑ | Spearman (Low MSA depth) ↑ | Spearman (Medium MSA depth) ↑ | Spearman (High MSA depth) ↑ |
> > | :---- | :---- | :---- | :---- | :---- |
> > | ESM2 (35M) | 0.3211 | 0.2394 | 0.2707 | 0.4510 |
> > | ESM2 (35M) \+ TTT\_MSA | 0.3366 | **0.3377** | **0.3430** | 0.3809 |
> > | ESM2 (35M) \+ TTT | 0.3406 | 0.2445 | 0.3142 | **0.4598** |
> > | ESM2 (35M) \+ TTT \+ TTT\_MSA | **0.3601** | **0.3377** | **0.3430** | **0.4598** |
> >
> >  *Please note that for this experiment we only used one random seed in the limited time frame of the rebuttal, while TTT is stable across different seeds, as shown in other experiments (e.g. Table 1).*
> >
> >
> >  **TTT can be used to boost performance of models trained on MSAs.** Additionally, we implement MSATransformer+TTT, applying TTT to MSATransformer ([Rao et al., 2021](https://www.biorxiv.org/content/10.1101/2021.02.12.430858v3)). Specifically, we apply the same TTT objective function used for a single test sequence, while also incorporating the MSA as input to the model, as done in the MSA Transformer. The table below (not yet incorporated into the manuscript) shows that, even without tuning hyperparameters (reused from experiments with ESM2), MSATransformer+TTT outperforms MSATransformer.
> >
> > | Model | Avg. Spearman ↑ |
> > | :---- | :---- |
> > | MSA Transformer | 0.4206 |
> > | MSA Transformer \+ TTT | **0.4223** |
> >
> >  *Please note that for this experiment we only used one random seed in the limited time frame of the rebuttal, while TTT is stable across different seeds, as shown in other experiments (e.g. Table 1). Please also note that for the same reason we do not use an ensemble with 5 random seeds as done in ProteinGym.*

---

> > > ### Author Response · Authors · 2024-11-22
> > >
> > > > **Q1.** In Table 1, is the TTT done per protein individually or simultaneously for all proteins in the test set? Does the same applies to the other benchmarked methods as well?
> > >
> > > We thank the reviewer for highlighting this important point. In all our experiments, TTT is applied per protein, which is a key aspect of our method. This approach allows customizing (or fine-tuning) the protein language model to one specific target test sequence by self-supervised masked learning on that sequence (without seeing any other sequences). The intuition is that the weights of the model are slightly shifted towards the testing sequence which effectively allows the model to **specialize to** that specific target test sequence. This capability to specialize may not be attainable through fine-tuning on larger datasets. We have clarified it in the captions of all main tables: Table 1, Table 2, and Table 3\.
> > >
> > > > **Q2.** Since TTT on one test protein is motivated by the derivation of MSAs to be time consuming, have you tested variants of ESM2 and SaProt where the models (fog) are finetuned by the homologous sequences derived for each protein? Do these variants outperform TTT fine tuned models? My understanding is that ESM2 and SaProt see similar proteins to the test/target protein in their training but are not further fine tuned on the MSA derived per protein.
> > > (i) Also, it would have been nice to see a combination of MSA and TTT, where TTT is done for all the homologous sequences to the test protein. It is not clear to me whether it hurts or improves the performance compared to only using TTT on the single test protein.
> > > (ii) The point of homologous sequences can be applied to the other included benchmarks too.
> > >
> > > We believe that our response to **W4** addresses the questions by discussing the new TTT\_MSA approach, which (i) combines TTT with MSA to improve performance and (ii) can be applied to all the benchmarks discussed in the paper.
> > >
> > > > **Q3.** What is meant by TTS steps?
> > >
> > > We could not find the term "TTS steps" in the text of our paper. Nonetheless, if the question is about TTT steps, TTT stands for Test-Time Training (Section 3). We also use the acronym TPS for TerPene Synthase protein (Section 4.3).
> > >
> > > > **Q4.** Is there any preference on what h (task specific head) has been trained on? Should it be the same dataset as the one used for backbone (f) training, e.g., like the structure prediction task using ESMFold (Section A.2.3), or it could be any other large dataset with information on the task.
> > >
> > > We thank the reviewer for highlighting this point. The motivation for considering different tasks (protein fitness, structure, and function prediction) and different models was to demonstrate the flexibility of our TTT method. Our results suggest that TTT is robust to both the different downstream task heads $h$ and the datasets used to train them. For instance, the ESMFold head is a large AlphaFold 2-inspired model with more than a billion parameters trained on PDB and AlphaFold DB, whereas the TPSMiner head is a compact random forest trained on a limited dataset of hundreds of terpene synthase enzymes.

---

### Author Response · Authors · 2024-11-22

We thank the reviewers for their constructive and insightful feedback. Below we summarize the main strengths of our paper as they have been pointed out by the reviewers as well as how we address the main weaknesses.

**Strengths**

- The proposed technique has been extensively benchmarked (UQtJ) and achieves new state-of-the-art results on three tasks (dg8u). The experiments are of reasonable quality with statistically significant results (dg8u).
- The paper applied the proposed technique for the first time to the computational problems in the protein domain (UQtJ) and can positively influence the future work in this area (dg8u).
- The paper is well-written and easy to follow (dg8u, Rmaa).
- The paper is well-motivated (dg8u, Rmaa) and has a nice related work section (dg8u).
- The appendix contains valuable and complementary information to the paper (dg8u).

**Weaknesses**

- **Test-time training (TTT) proposed in this work may underperform multiple sequence alignment (MSA) (UQtJ, Rmaa)**. We show that TTT is not intended to compete with MSA but rather serves as an orthogonal complementary technique. We have conducted additional experiments to demonstrate that TTT can be combined with MSA (responses to UQtJ W3-W4, Rmaa W2) or applied in scenarios where MSA is unavailable or costly to compute (response to Rmaa W3).
- **Hyperparameter optimization for TTT (dg8u)**. We have clarified our approach to estimating TTT hyperparameters, emphasizing two key points (response to dg8u W1). First, while estimating the number of TTT steps is important, we discussed that the expected performance of TTT can be controlled by tuning the learning rate and batch size while keeping the number of steps constant. Second, we highlighted the relationships between the number of fine-tuned parameters and the performance of TTT emerging from our results.
- **The choice of benchmarks and baselines (Rmaa).** We have added additional clarifications that in our work we have chosen well-established benchmarks and baselines across different problems to demonstrate the broad potential of TTT. We have also conducted additional experiments (response to Rmaa W3) and analyses (response to Rmaa W1) justifying our choices.

The changes to the manuscript reflecting our responses are highlighted in blue in the updated pdf. In order to get the response out now, **we have not yet included in the updated pdf of the submission all the new results presented in the responses below**. However, we will include all the responses and new results in the camera-ready version, should the paper be accepted.

---

> ### Author Response · Authors · 2024-12-02
>
> Dear Reviewers, we appreciate your time and thoughtful feedback. As the discussion period is nearing its conclusion, we kindly invite you to review our responses at your convenience (if you have not done so already). If there are any points that remain unclear or require further elaboration, we would be more than happy to provide additional clarifications promptly. We also hope that our responses have addressed your concerns. Thank you again for your time, effort, pointers to related work and invaluable insights.

---

### Meta-Review · Area_Chair_UmPX · 2024-12-21

**Metareview:**

This paper applies training/fine tuning at test time (TTT) to update the representations learned by foundation models on a large corpus of protein families, to a specific task defined on a test protein.
The reviewers emphasized that the paper is well-written and shows strong results. However, the method is not new, only its application in the context of proteins. Furthermore, the reviewers questioned theoretical grounding, remarked upon the cost of fine-tuning and the lack of a direct comparison with established methods like TranceptEVE. These last points are the most important in the decision to reject.

**Additional Comments On Reviewer Discussion:**

Many points were raised and addressed during the rebuttal. The points on the lack of theory and the lack of a direct comparison with established methods were not fully resolved.

---

### Decision · Program_Chairs · 2025-01-22

Reject